# BUCKINGHAM $\pi$-INVARIANT TEST-TIME PROJECTION FOR ROBUST PDE SURROGATE MODELING

**Seokki Lee**[1*], **Min-Chul Park**[2*], **Giyong Hong**[1*], **Changwook Jeong**[1,3†]
[1]Graduate School of Semiconductor Materials and Devices Engineering, UNIST
[2]Computational Science and Engineering Team, Device Solutions, Samsung Electronics
[3]AI Graduate School, UNIST
{seokki, e150great, changwook.jeong}@unist.ac.kr
m.c.park@samsung.com

## ABSTRACT

PDE surrogate models such as FNO and PINN struggle to predict solutions across inputs with diverse physical units and scales, limiting their out-of-distribution (OOD) generalization. We propose a $\pi$-invariant test-time projection that aligns test inputs with the training distribution by solving a log-space least squares problem that preserves Buckingham $\pi$-invariants. For PDEs with multidimensional spatial fields, we use geometric representative $\pi$-values to compute distances and project inputs, overcoming degeneracy and singular points that limit prior $\pi$-methods. To accelerate projection, we cluster the training set into $K$ clusters, reducing the complexity from $\mathcal{O}(MN)$ to $\mathcal{O}(KN)$ for the $M$ training and $N$ test samples. Across wide input scale ranges, tests on 2D thermal conduction and linear elasticity achieve MAE reduction of up to $\approx 91\%$ with minimal overhead. This training-free, model-agnostic method is expected to apply to more diverse PDE-based simulations.

## 1 INTRODUCTION

A central principle of physics is *dimensional homogeneity*: every valid physical law can be expressed in terms of dimensionless groups, known as *Buckingham $\pi$-invariants*. Once recast in terms of these invariants, a system's behavior remains unchanged under any rescaling of dimensional variables that preserves the values of the $\pi$-groups. For instance, flows in a narrow pipe and in a vast river are dynamically equivalent when their Reynolds number matches. This implies that certain apparent out-of-distribution (OOD) shifts, often regarded as a major challenge in physics-informed machine learning and neural operator models, may not constitute genuine distribution shifts at all, but merely *scale changes* that are physically equivalent under the Buckingham $\pi$ framework.

Building on this observation, we propose a general, training-free, and model-agnostic test-time procedure, *$\pi$-invariant projection*. Given the input fields and parameters of a partial differential equation (PDE), we map each test sample into the neighborhood of the training distribution *while preserving its $\pi$-values*: specifically, we move the sample within its own *$\pi$-equivalence class* and project it onto the nearest training $\pi$-equivalence class. As a result, inputs are aligned according to their essential physical similarity rather than arbitrary scale. The framework is applicable to a broad class of PDEs wherever dimensional analysis identifies invariant groups, and integrates as a drop-in inference step for any surrogate model, including U-Nets and neural operators. To reduce computational cost, we replace exhaustive pairwise comparisons with a centroid scheme: clustering $M$ training samples into $K$ representatives yields $\mathcal{O}(KN)$ test-time complexity instead of $\mathcal{O}(MN)$ for $N$ test instances. Across diverse physical simulations, $\pi$-invariant projection substantially improves OOD generalization, often approaching in-distribution accuracy.

---

*Equal contribution.
†Corresponding author.

## 2    RELATED WORKS

**Fourier Neural Operator (FNO).**    The Fourier neural operator (FNO) is a resolution-agnostic surrogate that learns mappings in Fourier space, enabling resolution-independent predictions (Li et al., 2020). Numerous FNO variants followed (Bartolucci et al., 2023; Kossaifi et al., 2023; Rahman et al., 2022). However, truncating to a limited set of low-frequency modes can discard physically relevant high-frequency content; Physics-Informed Neural Operators (PINO) mitigate this by adding PDE-constrained losses to recover high-frequency behavior (Li et al., 2024). Overall, FNO excels at in-distribution interpolation but struggles to extrapolate OOD when inputs have disparate units and scales.

**Dimensionless learning.**    Learning dimensionless $\pi$-groups reduces variables and enforces scale invariance, enabling scalar extrapolation with neural networks or SINDy (Bakarji et al., 2022; Xie et al., 2022; Oppenheimer et al., 2023; Gunaratnam et al., 2003; Brunton et al., 2016). However, most work focuses on scalars rather than spatial fields. For fields, naive Buckingham–$\pi$ scaling has two failures: (i) numerator zeros collapse many inputs to zero; and (ii) near-zero denominators make $\pi$ undefined or unbounded, destabilizing learning and analysis. These issues are rarely addressed, limiting direct use in 2D/3D. Recent efforts couple $\pi$-groups with nonlinear maps in turbulent flows (Fukami et al., 2024), but a systematic treatment of zero-set collapse and denominator blow-ups remains missing.

**Generalization for the Out-of-distribution (OOD) data.**    Uncertainty methods estimate OOD error but seldom improve accuracy without fine-tuning (Gal & Ghahramani, 2016; Lakshminarayanan et al., 2017; Fuchsgruber et al., 2024; Angelopoulos et al., 2024). Test-time training/adaptation updates a pretrained model during inference to boost OOD performance (Gandelsman et al., 2022; Wang et al., 2020; Adachi et al., 2024), but adds optimization overhead and latency and remains rare for regression and spatial fields. A promising alternative is sample-wise, $\pi$-preserving test-time alignment without TTT, which is underexplored.

**Our Contributions.**    This work addresses inefficiency and generalization challenges in PDE-based physical prediction caused by differences in input units and ranges, by introducing a test-time projection based on Buckingham $\pi$-invariance. The projection maps test samples near the training samples while preserving a governing dimensionless group, and a centroid-based scheme reduces computational complexity from $\mathcal{O}(MN)$ to $\mathcal{O}(KN)$. The module is training-free and model-agnostic, significantly improving OOD performance across various physical simulations, often approaching in-distribution accuracy. The key contributions are three-fold.

1. $\pi$-preserving test-time projection for spatial fields: an explicit, invariant input transform that maps each test sample toward the training distribution under the governing $\pi$, improving robustness to out-of-distribution shifts.

2. $\pi$-uniform strategy: uniformizes the sample-wise $\pi$ distribution by tuning the dominant-scale input while others fixed, enabling balanced training coverage which can be a general methodology in dimensionless modeling.

3. centroid reduction for projection: replaces exhaustive pairwise comparisons with centroid representatives, reducing test-time complexity from $\mathcal{O}(MN)$ to $\mathcal{O}(KN)$ with negligible accuracy impact.

Although our proposed framework can be applied to a wide range of PDE analyses, we focused on thermal and stress problems. Additional examples of applying the PDE are shown in App. F. In Section 3, we introduce the background of Buckingham-$\pi$ and define the dimensionless parameters in the thermal conduction and linear elasticity. Section 4, the method for model-agnostic $\pi$-invariant projection is explained. In addition, at the test-time, efficient method in terms of computation cost is introduced throughout the $\pi$-uniform strategy and $K$-means clustering. Section 5 describes the experimental procedure in detail, while Section 6 presents the corresponding results. Finally, Section 7 present a discussion on the experimental limitations and the conclusions.

## 3 BUCKINGHAM $\pi$ THEOREM

The Buckingham $\pi$-theorem provides a framework for dimensional analysis, transforming physical variables into dimensionless $\pi$-groups to reduce complexity and ensure scale invariance. If we know the units of parameters in the given PDE, the $\pi$-groups are easily extracted by eliminating the units of denominator and numerator. For a system with $n$ variables and $m$ fundamental units (e.g., mass, length, time), it yields $n - m$ dimensionless groups (see App.A). By recasting the problem in terms of these invariants, seemingly distinct inputs with different scales can be recognized as physically equivalent, providing a theoretical foundation for robust OOD generalization. Below, we define representative $\pi$-groups for thermal conduction and stress simulation, used to motivate invariants and scaling rules.

**Theorem 1** ((Buckingham $\pi$, log form)). *Given $B$ base units, $p$ dimensional variables $x \in \mathbb{R}_{>0}^p$ and a dimension matrix $D \in \mathbb{R}^{|B| \times p}$ (rows = base units, columns = variables), let $r = \mathrm{rank}(D)$. Then there exist $p - r$ independent dimensionless combinations ("$\pi$-groups"). If $\Phi \in \mathbb{R}^{p \times (p-r)}$ spans $\ker(D)$, then*

$$\log \Pi(x) = \Phi^\top \log x \quad \Longleftrightarrow \quad \Pi(x) = \exp(\Phi^\top \log x)$$

*is a complete set of $p - r$ independent $\pi$-groups. (See App. A for details.)*

**Notation.** $\Phi = [\phi^{(1)} \cdots \phi^{(p-r)}]$ stacks the null-space basis vectors; each column $\phi^{(\ell)}$ defines one dimensionless monomial (a $\pi$-group). The set of log-rescalings that preserve all $\pi$-values is $\ker(\Phi^\top) = \{ v \in \mathbb{R}^p : \Phi^\top v = 0 \}$.

**How scaling acts (log-space translation).** A componentwise rescaling $x \mapsto x \odot \exp(v)$ becomes a translation $z = \log x \mapsto z + v$ in log space. If $v \in \ker(\Phi^\top)$, then $\Phi^\top(z+v) = \Phi^\top z$, i.e., $\pi$-values are unchanged. Thus each input $z$ generates an affine $\pi$-*equivalence class* $z + \ker(\Phi^\top)$.

In short, the physics lives in the $\pi$-values: unit or scale changes are just $\pi$-preserving shifts in log space, so inputs related by such shifts are the same case.

**Worked example (thermal).** With base units $(M, L, T, \Theta)$ and variables $(k, q, \Delta T, L)$,

$$[k] = MLT^{-3}\Theta^{-1}, \quad [q] = ML^{-1}T^{-3}, \quad [\Delta T] = \Theta, \quad [L] = L.$$

, the dimension matrix $D$ can be formed.

$$D = \begin{bmatrix} 1 & 1 & 0 & 0 \\ 1 & -1 & 0 & 1 \\ -3 & -3 & 0 & 0 \\ -1 & 0 & 1 & 0 \end{bmatrix}, \qquad \text{columns: } \{k, q, \Delta T, L\}, \quad \text{rows: } \{M, L, T, \Theta\}. \tag{1}$$

In general, for a variable $x_j$ with units $\prod_{b \in B} b^{D_{b,j}}$, the $j$-th column of $D$ stores its exponents. The number of $\pi$ groups defined by the Buckingham $\pi$ theorem is $p - \mathrm{rank}(D)$. Because $\mathrm{rank}(D) = 3$, only one $\pi$ value is defined by the theorem.

A dimensionless monomial corresponds to a null-space vector $\phi \in \ker(\mathrm{D})$, i.e., $D\phi = 0$. Stacking independent solutions gives a $\pi$-basis $\Phi = [\phi^{(1)} \cdots \phi^{(p-r)}]$ with $D\Phi = 0$, but in this case $p - r = 1$, so one vector suffices. A $\pi$-basis (null-space basis) is $\Phi = [\ \phi^{(1)}\ ]$, where

$$\phi^{(1)} = \begin{bmatrix} -1 \\ 1 \\ -1 \\ 2 \end{bmatrix},$$

which produces

$$\pi_{\mathrm{th}} = k^{\phi_k} q^{\phi_q} (\Delta T)^{\phi_{\Delta T}} L^{\phi_L} = k^{-1} q^1 (\Delta T)^{-1} L^2 = \frac{q L^2}{k \Delta T}.$$

$(M, L, T, \Theta)$ are units of mass, length, time, and temperature, respectively. For the variables, $k$ is a thermal conductivity field with its unit of $[W/(mK)]$, $q$ is a volumetric heat source field with $[W/m^3]$, $\Delta T$ is a scalar value from the difference between maximum and minimum value of boundary conditions with $[K]$, and $L$ is the grid length with $[m]$. The $\pi_{th}$ is the constraint our projection enforces (thermal row of equation 12).

**Worked example (elasticity).** Elasticity proceeds analogously. With base units $(M, L, T)$ and variables $(E, \sigma, f, L, \Delta u)$, the dimension matrix $D$ can be formed.

$$D = \begin{bmatrix} 1 & 1 & 1 & 0 & 0 \\ -1 & -1 & -2 & 1 & 1 \\ -2 & -2 & -2 & 0 & 0 \end{bmatrix}, \qquad \text{columns: } \{E, \sigma, f, L, \Delta u\}, \quad \text{rows: } \{M, L, T\}. \qquad (2)$$

For the variables, $E$ is the Young's modulus field with its unit of $[Pa]$, $\sigma$ is a stress field with $[Pa]$, $f$ is a body force per volume field with $[N/m^3]$, $L$ is the grid length with $[m]$, and $\Delta u$ is a displacement value from the difference between maximum and minimum value of boundary conditions with $[m]$.

A $\pi$-basis (null-space basis) can be collected as

$$\Phi = \begin{bmatrix} \phi^{(1)} & \phi^{(2)} & \phi^{(3)} \end{bmatrix}, \qquad D\Phi = 0,$$

with columns

$$\phi^{(1)} = \begin{bmatrix} -1 \\ 1 \\ 0 \\ 0 \\ 0 \end{bmatrix} \Rightarrow \pi_1 = \sigma/E, \qquad \phi^{(2)} = \begin{bmatrix} -1 \\ 0 \\ 1 \\ 1 \\ 0 \end{bmatrix} \Rightarrow \pi_2 = fL/E, \qquad \phi^{(3)} = \begin{bmatrix} -1 \\ 0 \\ 1 \\ 0 \\ 1 \end{bmatrix} \Rightarrow \pi_3 = f\,\Delta u/E.$$

Details are in App. A for the task-specific $\pi$ and the resulting linear log-constraint.

## 4 METHOD

We propose a training-free, model-agnostic $\pi$-invariant projection[*] to align OOD inputs with training data while preserving Buckingham $\pi$-groups (Section 3). The pipeline has three stages: (1) *$\pi$-preserving test-time projection* which projects each test sample toward the training data sample while preserving the $\pi$ via dimension reduction, log-scale linearization, projection, and prediction; (2) *$\pi$-uniform strategy* that generate uniform $\log \pi$ distribution to cover the wide range of $\pi$ group, improving the model prediction accuracy; (3) *Centroid reduction for projection*: combined with the $\pi$-uniform strategy, we run K-means clustering on log-space training features and use the $K$ centroids as projection representatives.

### 4.1 PROBLEM DEFINITION

We are given a training dataset $\mathcal{D}_{\mathrm{tr}} = \{(X_i, Y_i)\}_{i=1}^{M}$, where each input $X_i$ consists of discretized input fields together with the associated physical parameters and discretization step, and each output $Y_i$ is the corresponding discretized solution of a governing PDE. All variables and field values are dimensional; see Fig. 1. This dataset is used to train a chosen surrogate PDE model. Then, for a given test input $\tilde{X}$, our goal is to construct a transformed input $\tilde{X}^*$ that (i) preserves the Buckingham–$\pi$ invariants of $\tilde{X}$ and (ii) lies close to the training distribution under a scale-appropriate metric.

More formally, let $\tilde{x} = \psi(\tilde{X})$ denote the vector of dimensional variables extracted from $\tilde{X}$, where $\psi$ is a feature extractor. For algebraic equations, $\psi(\cdot) = \mathrm{Id}(\cdot)$. For differential equations, we need to extract the representative variables from $\tilde{X}$ using $\psi$, which will be discussed later. Then, the problem is defined as

$$\tilde{x}^* = \underset{\tilde{x}' \in [\psi(\tilde{X})]_\pi}{\arg\min} \left[ \mathrm{dist}(\tilde{x}', \{\psi(X_i)\}_{i=1}^{M}) \right], \qquad (3)$$

where $\mathrm{dist}(\cdot)$ is a chosen distance measure between a candidate point and the training set.

---

[*]The algorithm of the Buckingham $\pi$-invariant projection is shown in App. H.

### 4.2 Domain Profile Reduction

For algebraic systems, $\pi$-groups can be extracted directly from variables by enforcing dimensional homogeneity in Theorem 1. However, applying Buckingham-$\pi$ analysis directly to high-dimensional spatial fields in PDEs presents a fundamental challenge. A naive pixel-wise approach often leads to $\pi$-degeneracy, as shown in Fig. 1; if a field value is zero (e.g., $q \to 0$) or near-zero at a specific location, the resulting $\pi$-value may vanish or diverge, causing numerical instability.

To overcome this issue, we introduce a feature extractor $\psi : X \mapsto x$ that maps each discretized field to a finite set of characteristic variables. Among possible choices, we adopt the arithmetic mean of the field values. Unlike pixel-wise evaluation, this global statistic is robust to local zero values and outliers, ensuring non-zero representative scales that enable stable log-linear projection. For non-spatial inputs, these characteristic variables are used directly.

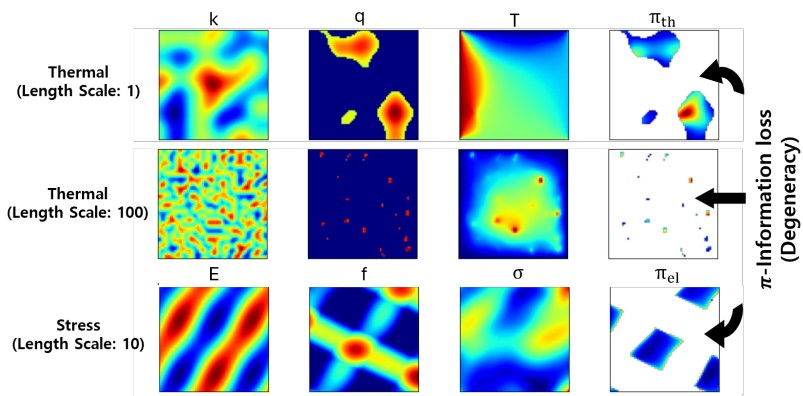

Figure 1: The PDE datasets for thermal and linear elasticity. With given distributions of $k, q, E, f$, the surrogate model predicts the fields of $T$ and $\sigma$ in each case (Thermal and Stress). Pixel-wise $\pi$ values go to zero when $q \to 0$ and $f \to 0$, which is described as $\pi$-information loss.

### 4.3 Offline Preparation: $\pi$-uniform strategy & Centroid reduction

The $\pi$-uniform strategy is adopted for enabling a balanced training coverage. It uniformizes the sample-wise $\pi$ distribution in train samples by tuning the dominant scale input (e.g., $q$ in thermal case) while others fixed. The dominant parameter is defined as the input parameter whose scale most directly controls the Buckingham $\pi$-group. Concretely, it can be done with SHAP analysis on group for each parameter. For example, the dominant parameter in thermal case shows the highest importance among whole parameter's contribution (48.7%). Once the dominant parameter is fixed, we first sample the input parameters from broad uniform ranges and compute their $\pi$-values. From this $\pi$-distribution, we define a target uniform distribution in $\pi$-space; each training sample adjusts only the dominant parameter (e.g., $q$ for thermal). so that its $\pi$-value matches the target while other inputs fixed. By making $K$ clusters in uniformized $\pi$ distribution, the centers of $K$ clusters represent whole distribution. In our approach, the distribution of $\log \pi$ is uniformized rather than that of $\pi$ for enlarging the coverage. Naïvely, comparing $N$ tests to all $M$ training samples entails $\mathcal{O}(MN)$ pairwise evaluations of the inter-class residual in equation 3. To reduce this cost, we cluster training log-features with $K$-means and keep $K$ centroids, cutting complexity to $\mathcal{O}(KN)$ while preserving projection accuracy (App. D; Sec. 5.5). (see App. C)

### 4.4 Online Test-Time Projection

The main goal is to find the optimal projection, i.e., the closest point between the *equivalence class* generated by the test point $\tilde{X}$ and *equivalence classes* generated by each train data $(X_i, Y_i)$. Since the vector $v$ that generates equivalence classes in log-space is simply a parallel translation, the space it spans is of the form $z + \ker(\Phi^\top)$. This implies that the entire space decomposes into two parts: (i) the component perpendicular to $\ker(\Phi^\top)$ (inter-class variation, i.e., the physical change), and

(ii) the component parallel to $\ker(\Phi^\top)$ (intra-class change, i.e., the scaling variation). Ultimately, we identify the closest training class to the test class in the inter-class direction, and then adjust the test sample within its own equivalence class to achieve a scale consistent with the chosen training samples.

More formally, setting $z_i = \log x_i$ and $\tilde{z} = \log \tilde{x}$, the optimization problem equation 3 becomes

$$\min_{\tilde{z}' \in \tilde{z} + \ker \Phi^\top} \mathrm{dist}\left(\tilde{z}', \{z_i\}_{i=1}^M\right) = \min_{i=1,\ldots,M} \min_{v \in \ker \Phi^\top} \|z_i - (\tilde{z}+v)\|_2 = \min_{i=1,\ldots,M} d_{\sim\pi}([x_i],[\tilde{x}]), \quad (4)$$

where $i$ indexes the training samples, $[\tilde{x}]_\pi$ and $[x_i]_\pi$ denote the *equivalence classes* of the test and $i$-th training samples, respectively. Thus, the problem reduces to finding the nearest training class in terms of *the quotient distance* between the fixed $[\tilde{x}]_\pi$ and the union of the training equivalence classes $\bigcup_{i=1}^M [x_i]_\pi$. See App. B for details on the used metric.

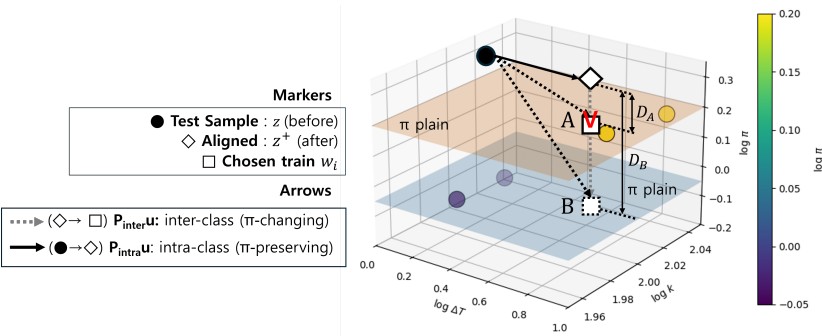

Figure 2: The schematic of $\pi$-invariant projection; the test sample (marked as circle) projects to the optimal train sample $x_A$ ($\because D_A < D_B$).

**Step 1. Decomposition of optimal scaling vector**

Let $v_i^t = z_i - \tilde{z} = \log(x_i/\tilde{x})$ in equation 4 denote the unconstrained log-scaling factor between $x_i$ and $\tilde{x}$. Each $v_i^t$ can be decomposed into *intra-class* (scaling) and *inter-class* (physics) parts using the projector onto $\ker(\Phi^\top)$:

$$P_\| = I - \Phi(\Phi^\top\Phi)^{-1}\Phi^\top \quad \left(\text{or } P_\| = I - \Sigma^{-1}\Phi(\Phi^\top\Sigma^{-1}\Phi)^{-1}\Phi^\top \text{ with a weight } \Sigma\right),$$
$$v_i^t = \underbrace{P_\| v_i^t}_{\text{intra-class (scales, } \pi\text{-preserving)}} + \underbrace{(I - P_\|)v_i^t}_{\text{inter-class (changes } \pi)} \quad (5)$$

As an example, Figure 2 illustrates $v_1^t$ and $v_2^t$, which are defined from the test sample $\tilde{x}$ and the two training candidates $x_1$ and $x_2$. For each $i$, choosing $v_i^* = P_\| v_i^t$ yields the quotient distance between $[x_i]$ and $[\tilde{x}]$, as the projection aligns the training and test samples in their respective affine spaces, isolating the class-wise irreducible component of the distance $(I - P_\|)v_i^t$.

**Step 2. Optimum train sample to be projected**

The optimum equivalence class to be projected has a minimum value of $\|v_i^t - v_i^*\|_2$ among whole training equivalence classes. The optimization problem can be represented as below:

$$i^* = \underset{i=1,\ldots,M}{\arg\min} \|v_i^t - v_i^*\|_2, \quad (6)$$

where $i$ represents the equivalence class number to be projected. For example, in the Figure 2, $\tilde{x}_1^*$ is the optimum point because $D_A = \|v_1^t - v_1^*\|_2 < D_B = \|v_2^t - v_2^*\|_2$. After that, the optimally scaled test sample is finally given by

$$\tilde{x}^* = \exp(\tilde{z}^*) = \exp(\tilde{z} + v^*) = \tilde{x} \odot \exp(v^*).$$

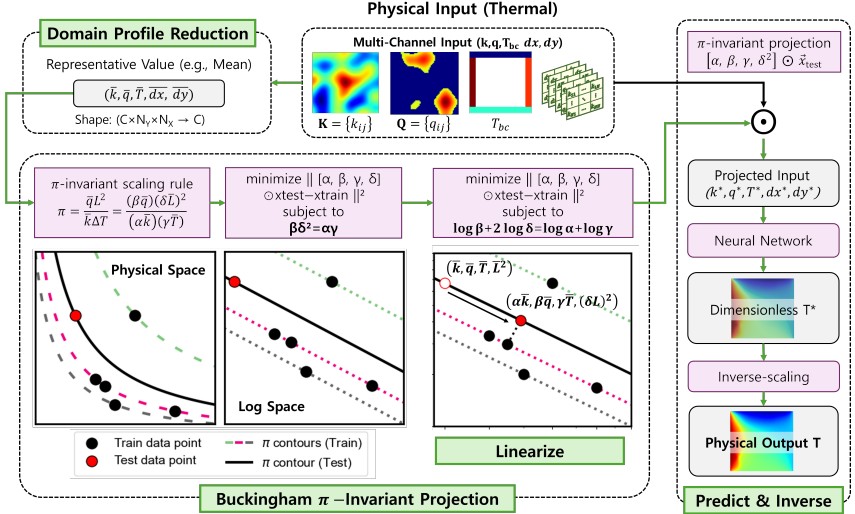

Figure 3: Flowchart of Buckingham $\pi$-Invariant Projection (Thermal case).

## 5 EXPERIMENTAL SETUP

The experiment proceeds in the following order: Training (Sec. 5.1) $\rightarrow$ Domain Profile Reduction (Sec. 5.2) $\rightarrow$ $\pi$-invariant Projection (Sec. 5.3) $\rightarrow$ Prediction (Sec. 5.4) — to perform OOD data prediction (Figure 3). In addition, for more efficient $\pi$-invariant projection, the $\pi$-uniform strategy and centroid reduction method can also be employed (Section 5.5).

### 5.1 TRAINING

**Thermal conduction model**  We solve the steady 2D conduction problem with Dirichlet boundary conditions on $T_{bc}$:

$$-\nabla \cdot (k\nabla T) = q. \tag{7}$$

We use three surrogate architectures: a plain CNN, U-Net, and FNO. Training and test sets are generated by numerically solving equation 7 on uniform grids, but with *disjoint parameter ranges* to induce OOD shift. For example, $\log_{10} q \in [0, 7.5]$ for training and $\log_{10} q \in [7.5, 12]$ for testing. Distributions of $k$, $q$, $T_{bc}$, and $dx$ are shown in Figure 4 (thermal panels). we encode this input into five channels $(k, q, T_{bc}, dx, dy)$.

**Linear elasticity model**  We consider plane stress/strain in 2D:

$$-\nabla \cdot \sigma = f, \quad \sigma = \mathbb{C}(E, \nu) : \varepsilon(u). \tag{8}$$

As in the thermal case, we evaluate CNN, U-Net, and FNO surrogates. Inputs are field maps $E$, $\nu$, and $f$, the displacement boundary condition $u_{bc}$, and grid spacings $dx, dy$. We encode $f$ and $u_{bc}$ via components $(f_x, f_y, u_{bc}^x, u_{bc}^y)$, yielding eight input channels: $(E, f_x, f_y, u_{bc}^x, u_{bc}^y, \nu, dx, dy)$. Parameter distributions for $(E, f, u_{bc}, dx)$ are shown in Figure 4 (elasticity panels).

### 5.2 DOMAIN PROFILE REDUCTION

To represent global characteristics of the domain, arithmetic mean values of input fields are adopted. As defined in Section 4.2, fields of $k$, $q$ (thermal) and $E$, $f$ (elasticity) are summarized to representative values $\bar{k}$, $\bar{q}$, $\bar{E}$, $\bar{f}$ respectively. However, representive $f$ ($\bar{f}$) may be extracted in a slightly different manner, because it is represented as two channels corresponding to $f_x$, $f_y$ components. Therefore, in this case we first compute the means $\bar{f}_x$, $\bar{f}_y$ of $f_x$, $f_y$, respectively, and then compute $\bar{f}$ using the $L_2$-norms of $\bar{f}_x$, $\bar{f}_y$ ($\bar{f} = \sqrt{\bar{f}_x^2 + \bar{f}_y^2}$). we set $dx = dy = L$ unless otherwise stated. For quantities that have zero values are not considered. The resulting feature vectors $\tilde{x}$ are shown in the pipeline (Figure 3).

### 5.3 $\pi$-INVARIANT PROJECTION

Prior to projection, the scaling process is essential in projection, enabling the surrogate model to relocate OOD onto an interpolative (in-distribution) scale and reliable predictions. For the thermal case, the scaling coefficients $\alpha, \beta, \gamma$ and $\delta$ are used to scale the parameters while preserving the $\pi_{\text{th}}$. Log-scale linearization is applied for efficient calculation (Section 3).

$$\bar{k}' = \alpha\bar{k}, \quad \bar{q}' = \beta\bar{q}, \quad \Delta T' = \gamma\Delta T, \quad L' = \delta L \tag{9}$$

For preserving the $\pi_{\text{th}}$, scaling coefficients should follow the constraint.

$$\pi_{\text{th}} = \frac{\bar{q}'L'^2}{\bar{k}'\Delta T'} = \frac{(\beta\bar{q})(\delta L)^2}{(\alpha\bar{k})(\gamma\Delta T)} = \frac{\bar{q}L^2}{\bar{k}\Delta T} \tag{10}$$

Likewise, scaling coefficients are applied to each input in stress case as below:

$$\bar{E}' = \alpha\bar{E}, \quad \bar{f}' = \beta\bar{f}, \quad L' = \gamma L$$

$$\pi_{\text{el}} = \frac{\bar{f}'L'}{\bar{E}'} = \frac{(\beta\bar{f})(\gamma L)}{(\alpha\bar{E})} = \frac{\bar{f}L}{\bar{E}} \tag{11}$$

Log formats are used to equations 25, 11 linearly transform scales of parameters.

$$\begin{cases} \log\beta + 2\log\delta - \log\alpha - \log\gamma = 0 \quad \text{(thermal)} \\ \log\beta + \log\gamma - \log\alpha = 0 \quad \text{(elasticity)} \end{cases} \tag{12}$$

which preserves $\pi_{\text{th}}$, $\pi_{\text{el}}$ in Appendix A.1, A.2. After that, inputs $[k, q, \Delta T, L]$ of given test sample $\tilde{x}$ are scaled by $\exp(v^*) = [\alpha, \beta, \gamma, \delta]$, and moved to the optimal projection point $\tilde{x}^*$ having inputs $\tilde{x}^* = \tilde{x} \odot \exp(v^*) = [\alpha k, \beta q, \gamma\Delta T, \delta L]$ from calculating the nearest class with $\pi$-preserving correction described in Section 4.4. The same rule is applied in the calculation of the elasticity.

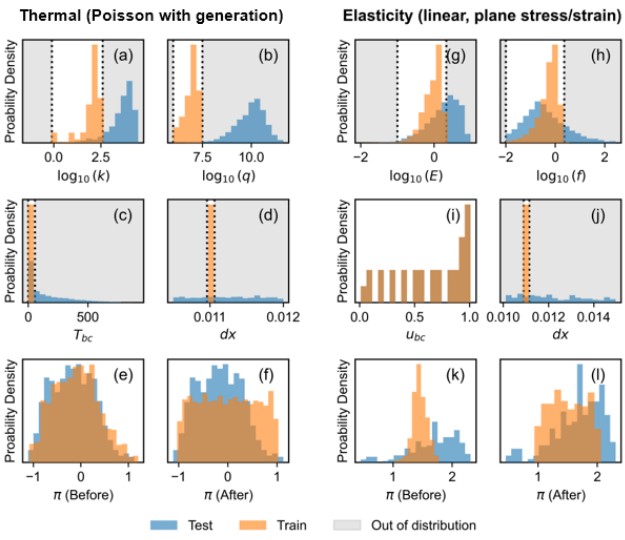

Figure 4: Histogram comparison for train (blue) and test (orange). Thermal (a–f): (a) $\log_{10} k$, (b) $\log_{10} q$, (c) $T_{bc}$, (d) $dx$; (e) $\log_{10} \pi_{\text{th}}$ before and (f) after $\pi$-uniform sampling. Elasticity (g-l): (g) $\log_{10} E$, (h) $\log_{10} \|f\|$, (i) $u_{bc}$, (j) $dx$, (k,l) $\log_{10} \pi_{\text{el}}$ (before vs. after).

### 5.4 PREDICTION PROCESS (SCALING, INVERSE SCALING)

After computing the optimal $v^*$, we rescale the test input channel-wise $\tilde{X}^* = \tilde{X} \odot \exp(v^*)$, i.e.,

$$\tilde{X}^* = \begin{cases} [\alpha k, \beta q, \gamma T_{bc}, \delta dx, \delta dy] & \text{(thermal)} \\ [\alpha E, \beta f_x, \beta f_y, \gamma u_{bc}^x, \gamma u_{bc}^y, \nu, \delta dx, \delta dy] & \text{(elasticity)} \end{cases} \tag{13}$$

Feeding $\tilde{X}^*$ to the surrogate model predicts $T_{\text{scaled}}$, and we invert the temperature scaling using the $\Delta T$ scale factor $\gamma$: $T = T_{\min} + \gamma^{-1}(T_{\text{scaled}} - T_{\min})$, where $T_{\min}$ is the minimum Dirichlet boundary temperature. (Analogous inverse maps apply for elasticity outputs.)

## 5.5 Centroid Reduction for Projection

In calculation the nearest class in Section 5.3, We apply a $\pi$-uniform sampling strategy (Sec.4.3) to obtain an approximately uniform distribution of $\log \pi$ (Figure 4 (e,f), (k,l)). Prior to applying the strategy, the $\log \pi$ distribution is one-sided and skewed; afterwards it becomes approximately uniform, indicating that the training space covers a broader range of $\pi$ values. For thermal we vary $q$ to target desired $\pi_{\text{th}}$ values; for elasticity we analogously vary $f$ for $\pi_{\text{el}}$. After $\pi$-uniform sampling, we run $K$-means on the training log-features $z$ to obtain $K \in \{1, \ldots, 10\}$ centroids. Because $K$-means is stochastic, we average results over 10 seeds (for both clustering and test sets). We compare *Baseline* (nearest over all training samples), *Clustered* (nearest over $K$ centroids), and *Random* (nearest over $K$ random training samples) using MAE, RMSE, and wall-clock. Timing excludes surrogate forward passes and reports only the projection stage (candidate selection + solving equation 3).

## 6 Results and discussion (Thermal and Elasticity)

Figure 5 compares *Top-3 best* and *Top-3 worst* OOD cases for thermal and elasticity. Across both, $\pi$-projection substantially improves fidelity, with the largest gains in the worst cases where the raw surrogate fails to extrapolate. Table 1 reports MAE/MSE and projection time for each architecture[†]. With the Buckingham-$\pi$ projection, both thermal and stress simulations achieve lower MAE and RMSE errors, but the inference time increases. However, applying the cluster algorithm with $\pi$ significantly reduce the inference time, while achieving a level of accuracy comparable to that obtained when computing distances over all training data.

The centroid reduction algorithm as described in 5.5 has demonstrated the efficiency in terms of computation cost when predicting the temperature distribution without a significant degradation in accuracy. From Figure 6, it is shown that the MAE tends to converge with the baseline projection as the number of candidates (cluster centers and random samples) increases while making a gap with random projection. The clustered projection matches Baseline MAE within statistical noise while reducing projection time by $\sim 100\times$ (Figure 6(c)).

**Handling $\pi$-degeneracy.** When $\pi_{\text{th}}$ is ill-defined or ill-conditioned (e.g., $q \approx 0$ or $\Delta T \approx 0$), we drop the explicit $\pi$ constraint and solve the same log-space alignment on the non-degenerate channels (e.g., $k, \Delta T, dx, dy$ in thermal). This removes global scale mismatch while preserving spatial heterogeneity, which empirically restores OOD accuracy. When valid $\pi$-groups are available (e.g., advection–diffusion, Navier–Stokes), the same log-affine machinery applies with the corresponding constraint.

Table 1: Test scores on thermal and elasticity. The best scores are bolded.

| Method | Thermal | | | Stress | | |
|---|---|---|---|---|---|---|
| | MAE | RMSE | Time | MAE | RMSE | Time |
| CNN | 8.43±0.29 | 9.99±0.33 | - | 0.96±0.08 | 1.17±0.10 | - |
| CNN + Pairwise Projection | 2.63±0.04 | 3.24±0.05 | 100.31±2.06 | **0.53±0.04** | **0.71±0.08** | 73.58±0.59 |
| CNN + $\pi$-uniform + 10-Randoms | 1.85±0.14 | 2.30±0.17 | **1.75±0.02** | 0.62±0.05 | 0.86±0.08 | **1.31±0.04** |
| CNN + $\pi$-uniform + 10-Centroids | **1.79±0.08** | **2.23±0.09** | 1.80±0.02 | 0.60±0.05 | 0.84±0.08 | 1.36±0.04 |
| U-Net | 13.60±0.31 | 15.29±0.35 | - | 0.81±0.08 | 0.99±0.10 | - |
| U-Net + Pairwise Projection | 1.75±0.01 | 2.31±0.02 | 99.12±3.90 | **0.17±0.03** | **0.28±0.08** | 94.91±1.70 |
| U-Net + $\pi$-uniform + 10-Randoms | 1.20±0.11 | 1.56±0.13 | **2.25±0.08** | 0.23±0.03 | 0.40±0.08 | **1.50±0.05** |
| U-Net + $\pi$-uniform + 10-Centroids | **1.18±0.08** | **1.53±0.10** | 2.31±0.07 | 0.22±0.03 | 0.39±0.08 | 1.58±0.07 |
| FNO | 9.88±0.23 | 11.43±0.26 | - | 3.20±0.84 | 4.19±1.03 | - |
| FNO + Pairwise Projection | 1.38±0.02 | 1.74±0.02 | 151.44±3.16 | **0.28±0.03** | **0.42±0.08** | 94.02±1.10 |
| FNO + $\pi$-uniform + 10-Randoms | 1.29±0.12 | 1.65±0.14 | **2.24±0.04** | 0.34±0.03 | 0.54±0.08 | **1.45±0.03** |
| FNO + $\pi$-uniform + 10-Centroids | **1.25±0.10** | **1.60±0.11** | 2.31±0.06 | 0.33±0.03 | 0.53±0.08 | 1.54±0.04 |

## 7 Limitations and Conclusion

**Limitations.** Our $\pi$-invariant projection assumes PDE-governed behavior and may underperform in hybrid settings with empirical components. Using representative statistics (e.g., arithmetic means

---

[†]We follow the authors' public implementation and adapt it to image-wise outputs; details in App. E.

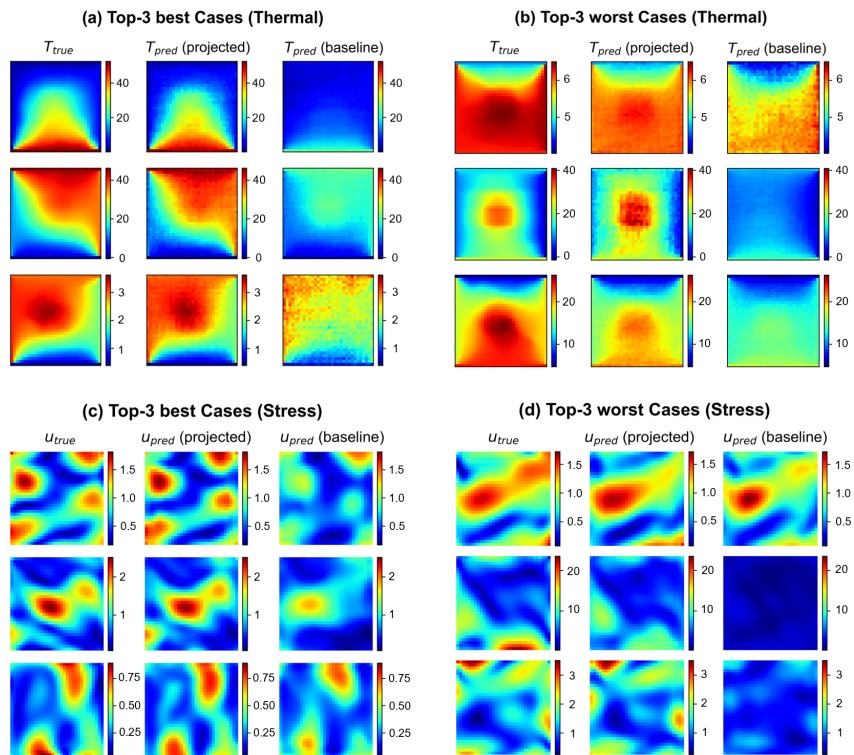

Figure 5: Comparison of true and predicted fields for *Top-3 best* (a) and *Top-3 worst* (b) cases in thermal (temperature) and elasticity (stress), under raw input vs. Buckingham $\pi$-invariant projection.

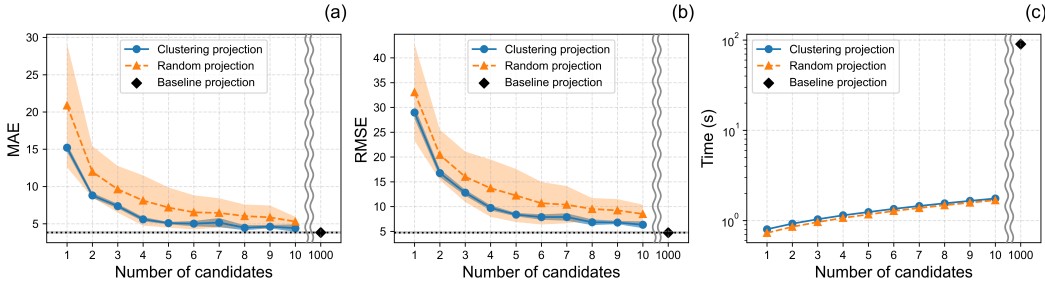

Figure 6: Performance comparison on (a) MAE, (b) RMSE, and (c) Time cost between clustering/random/baseline projection methods with 10 different test sets.

$\bar{k}$, $\bar{q}$) can blur highly irregular spatial patterns. While $\pi$-uniform sampling widens scale coverage, extreme OOD beyond the trained $\pi$ range still degrades accuracy (see App. G).

**Conclusion.** We introduce a compact, training-free, and model-agnostic $\pi$-invariant test-time projection that enforces physical similarity via Buckingham-$\pi$ and aligns OOD inputs to the training manifold through a tiny log-space least-squares scaling. This explicit log-scale transformation enables accurate predictions even under extreme out-of-distribution shifts. A centroid-based clustering variant accelerates inference, reducing complexity from $O(MN)$ to $O(KN)$ with negligible accuracy loss, and outperforms traditional test-time training in speed while maintaining high performance. On 2D thermal conduction and linear elasticity, the method reduces MAE by up to $\approx 91\%$ with minimal overhead. Future work includes uncertainty-aware projections, extensions to transient and convective PDEs (e.g., advection–diffusion, Navier–Stokes), and hybrid settings that incorporate empirical components.

ACKNOWLEDGMENTS

The authors thank In Huh for his valuable feedback on the theoretical presentation of this work, particularly regarding the group-theoretic interpretation of the quotient space and the geometric view of intra-/inter-class decomposition via projection operators. Changwook Jeong acknowledges support from Samsung Electronics (IO250618-13097-01); the Samsung Research Funding & Incubation Center for Future Technology of Samsung Electronics (SRFC-IT2502-01); the KIAT grant funded by MOTIE, Korea (P0023703, HRD Program for Industrial Innovation); and the IITP grant funded by MSIT, Korea (RS-2020-II201336, AIGS).

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

## A    DIMENSIONAL ANALYSIS VIA MATRIX NULL SPACE

By following the processes for each task, Buckingham $\pi$ groups can be defined.

### A.1    THERMAL CONDUCTION

Let the base dimensions be $B = \{M, L, T, \Theta\}$ (i.e. mass $M$, length $L$, time $T$, and temperature $\Theta$). We encode units as exponent vectors $x$ over $B$. For example, the unit of thermal conductivity ($k$) [W/(m $\cdot$ K)] is expressed as $x_k = [M^1 L^1 T^{-3} \Theta^{-1}]$, and taking a transpose, it is transformed into $x_k^\top = [1, 1, -3, -1]^\top$. Stacking the columns $x_k^\top, x_q^\top, x_{\Delta T}^\top, x_L^\top$ for the variables $[k, q, \Delta T, L]$ forms the dimension matrix $D \in \mathbb{R}^{|B| \times p}$ where $p$ is a total number of input and output parameters. The units and parameters are :

$k$ (thermal conductivity, [W/(m $\cdot$ K)]), $q$ (volumetric heat source, [W/m³]), $L$ (grid length, [m]), and $T$ (temperature, [K]).

$$D = \begin{bmatrix} 1 & 1 & 0 & 0 \\ 1 & -1 & 0 & 1 \\ -3 & -3 & 0 & 0 \\ -1 & 0 & 1 & 0 \end{bmatrix}, \qquad \text{columns: } \{k,\ q,\ \Delta T,\ L\}, \quad \text{rows: } \{M,\ L,\ T,\ \Theta\}. \qquad (14)$$

In general, for a variable $x_j$ with units $\prod_{b \in B} b^{D_{b,j}}$, the $j$-th column of $D$ stores its exponents. The number of $\pi$ groups defined by the Buckingham $\pi$ theorem is $p - \text{rank}(D)$. Because $\text{rank}(D) = 3$, only one $\pi$ value is defined by the theorem.

A dimensionless monomial corresponds to a null-space vector $\phi \in \ker(D)$, i.e., $D\phi = 0$. Stacking independent solutions gives a $\pi$-basis $\Phi = [\phi^{(1)} \cdots \phi^{(p-r)}]$ with $D\Phi = 0$, but in this case $p - r = 1$, so one vector suffices. A $\pi$-basis (null-space basis) is $\Phi = [\ \phi^{(1)}\ ]$, where

$$\phi^{(1)} = \begin{bmatrix} -1 \\ 1 \\ -1 \\ 2 \end{bmatrix},$$

which produces

$$\pi_{\text{th}} = k^{\phi_k} q^{\phi_q} (\Delta T)^{\phi_{\Delta T}} L^{\phi_L} = k^{-1} q^1 (\Delta T)^{-1} L^2 = \frac{q\, L^2}{k\, \Delta T}.$$

Unit check:

$$[\pi_{\text{th}}] = \frac{(\text{W/m}^3) \cdot (\text{m}^2)}{(\text{W}/(\text{m} \cdot \text{K})) \cdot (\text{K})} = \frac{\text{W/m}}{\text{W/m}} = 1,$$

so $\pi_{\text{th}}$ is dimensionless (any nonzero scalar multiple of $\phi$ yields the same $\pi$ up to a constant power).

### A.2    LINEAR ELASTICITY

We use base dimensions $B = \{M, L, T\}$ (no temperature). Let the variables be $(E, \sigma, f, L, \Delta u)$ with units $[E] = [\sigma] = M L^{-1} T^{-2}$, $[f] = M L^{-2} T^{-2}$, and $[L] = [\Delta u] = L$. The units and parameters are:

$E$ (Young's modulus, [Pa]), $f$ (body force per volume, [N/m³]), $L$ (grid length, [m]), $u$ (displacement, [m]), and $\sigma$ (stress, [Pa]).

The corresponding dimension matrix (rows $M, L, T$; columns as ordered) is

$$D = \begin{bmatrix} 1 & 1 & 1 & 0 & 0 \\ -1 & -1 & -2 & 1 & 1 \\ -2 & -2 & -2 & 0 & 0 \end{bmatrix}.$$

Here $p = 5$ and $\text{rank}(D) = 2$, so the nullity is $p - \text{rank}(D) = 3$, i.e., *three* independent $\pi$-groups.

A $\pi$-basis (null-space basis) is collected as

$$\Phi = [\ \phi^{(1)}\ \phi^{(2)}\ \phi^{(3)}\ ], \qquad D\Phi = 0,$$

with columns

$$\phi^{(1)} = \begin{bmatrix} -1 \\ 1 \\ 0 \\ 0 \\ 0 \end{bmatrix} \Rightarrow \pi_1 = \sigma/E, \qquad \phi^{(2)} = \begin{bmatrix} -1 \\ 0 \\ 1 \\ 1 \\ 0 \end{bmatrix} \Rightarrow \pi_2 = fL/E, \qquad \phi^{(3)} = \begin{bmatrix} -1 \\ 0 \\ 1 \\ 0 \\ 1 \end{bmatrix} \Rightarrow \pi_3 = f\,\Delta u/E.$$

(If grid spacings are anisotropic, replace $L$ by the arithmetic mean $L_{\mathrm{eff}} = \sqrt{dx\,dy}$; equivalently, use $\frac{1}{2}(\log dx + \log dy)$ in the log constraints.)

### A.2.1 MULTI $\pi$-INVARIANT PROJECTION

At test-time we enforce only *input-side* $\pi$-groups to avoid using unknown outputs. Thus we exclude $\pi_1 = \sigma/E$ and enforce $\pi_2 = fL/E$; if $\Delta u$ is provided as an input, we may additionally enforce $\pi_3 = f\,\Delta u/E$ for a tighter projection. Default: if $\Delta u$ is unavailable, enforce only $\pi_2 = fL/E$.

Let the input-side log-scales be $\mathbf{v} = [\log\alpha, \log\beta, \log\gamma, \log\delta]^\top$ acting on $E, f, L, \Delta u$ respectively, i.e.,

$$E' = \alpha E, \qquad f' = \beta f, \qquad L' = \gamma L, \qquad \Delta u' = \delta\,\Delta u.$$

Then the enforced $\pi$-constraints become linear equations in $\mathbf{v}$:

$$\underbrace{\begin{bmatrix} -1 & 1 & 1 & 0 \\ -1 & 1 & 0 & 1 \end{bmatrix}}_{C} \mathbf{v} = \mathbf{0} \quad\Longleftrightarrow\quad \begin{cases} \log\beta + \log\gamma - \log\alpha = 0 & (\pi_2 : fL/E), \\ \log\beta + \log\delta - \log\alpha = 0 & (\pi_3 : f\Delta u/E). \end{cases}$$

Given test log-features $\tilde{z}$ and a candidate train $z_i$, define $v_i^t = z_i - \tilde{z}$. Project $v_i^t$ onto the $\pi$-preserving subspace using either

$$\text{(i) projector onto } \ker(C): \quad P_\| = I - C^\top (CC^\top)^{-1} C \quad\Rightarrow\quad v^\star = P_\| v_{i^\star}^t$$

Or equivalently

$$\text{(ii) KKT solve for } \min_{\mathbf{v}} \|\mathbf{v} - v_i^t\|_2^2 \text{ s.t. } C\mathbf{v} = 0.$$

**Step 1 (nearest class under multiple $\pi$'s).**

$$i^\star = \arg\min_i \left\| (I - P_\|)\, v_i^t \right\|_2. \tag{15}$$

**Step 2 ($\pi$-preserving correction).** Apply $v^\star = P_\| v_{i^\star}^t$ and rescale channels by $(\alpha, \beta, \gamma, \delta) = \exp(\mathbf{v}^\star)$ to obtain the projected input. Run the surrogate on the projected input and inverse-scale the outputs (details in App. B).

**Notes.** (i) If only one of $\{L, \Delta u\}$ is present, the system reduces to the single-constraint case (enforce $fL/E$ or $f\Delta u/E$).
(ii) Under anisotropic discretization, replace the $L$-row in $C$ by $\frac{1}{2}[\cdots,\ \log dx + \log dy\,]$ (i.e., use $L_{\mathrm{eff}} = \sqrt{dx\,dy}$).
(iii) If a constraint becomes ill-conditioned (e.g., $f \approx 0$), drop that row of $C$ and project with the remaining valid constraints (degeneracy fallback).

## B INPUT-SPACE L2 METRIC ON LOG-SCALE

**Definition 1** (Logarithmic distance). *For $x, y \in \mathbb{R}^p_{>0}$ with logarithmic coordinates $z = \log x$ and $w = \log y$, define the distance*

$$d(x, y) = \|\log x - \log y\|_2 = \|z - w\|_2.$$

*Then, $d$ is a metric on $\mathbb{R}^p_{>0}$, because $x \mapsto \log x$ is injective on $x \in \mathbb{R}^p_{>0}$ and $\|\cdot\|_2$ is the standard Euclidean norm.*

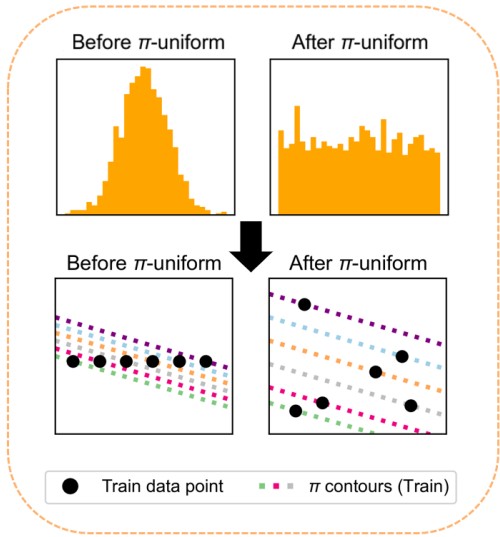

Figure 7: The schematic of $\pi$-uniform strategy

**Remark 1.** *Under the multiplicative scaling action as in Section 3, we have*

$$d(\rho(v,x), \rho(v,y)) = \|(z+v) - (w+v)\|_2 = \|z-w\|_2 = d(x,y),$$

*Hence, the metric $d$ is invariant under this group action.*

**Remark 2.** *The metric $d$ induces the standard point-to-set distance from a point $w = \log y$ to the $\pi$-equivalence class $[x]_\pi$ (or equivalently $[z]_\pi = \mathcal{E}(z)$) of a fixed $z = \log x$ as*

$$\operatorname{dist}(y, [x]_\pi) = \operatorname{dist}(w, \mathcal{E}(z)) = \inf_{v \in \ker \Phi^\top} \|w - (z+v)\|_2.$$

*Then, define the class-to-class (quotient) distance by*

$$\operatorname{dist}([y]_\pi, [x]_\pi) = \operatorname{dist}(\mathcal{E}(w), \mathcal{E}(z)) = \inf_{u,v \in \ker \Phi^\top} \|(w+u) - (z+v)\|_2.$$

*Since $d$ is invariant under translations by $-u \in \ker(\Phi^\top)$, this reduces to the point-to-class distance:*

$$\operatorname{dist}\big(\mathcal{E}(w), \mathcal{E}(z)\big) = \inf_{u,v \in \ker \Phi^\top} \|w - (z+v-u)\|_2 = \inf_{s \in \ker \Phi^\top} \|w - (z+s)\|_2 = \operatorname{dist}(w, \mathcal{E}(z)).$$

## C  $\pi$-UNIFORM STRATEGY

Figure 7 shows the schematic of $\pi$-uniform strategy. The dense $\pi$-contours in train sample are distributed uniformly by tuning dominant-scale input parameters (e.g., $\bar{q}$, $\bar{f}$ in thermal and elasticity case, respectively) so that the set of centroids can cover the distribution of all train samples when clustering is performed.

## D  CLUSTERING PROJECTION

The clustering projection proceeds as follows: (1) Uniformly adjust the pi distribution of the train samples using the $\pi$-uniform strategy. (2) Perform K-means clustering based on the $\pi$ values and determine the centroid for each cluster (marked with a star symbol). (3) Find the closest centroid for a given test sample and project the test sample to the closest position on the $\pi_{test}$ plane ($\pi$-invariant projection).

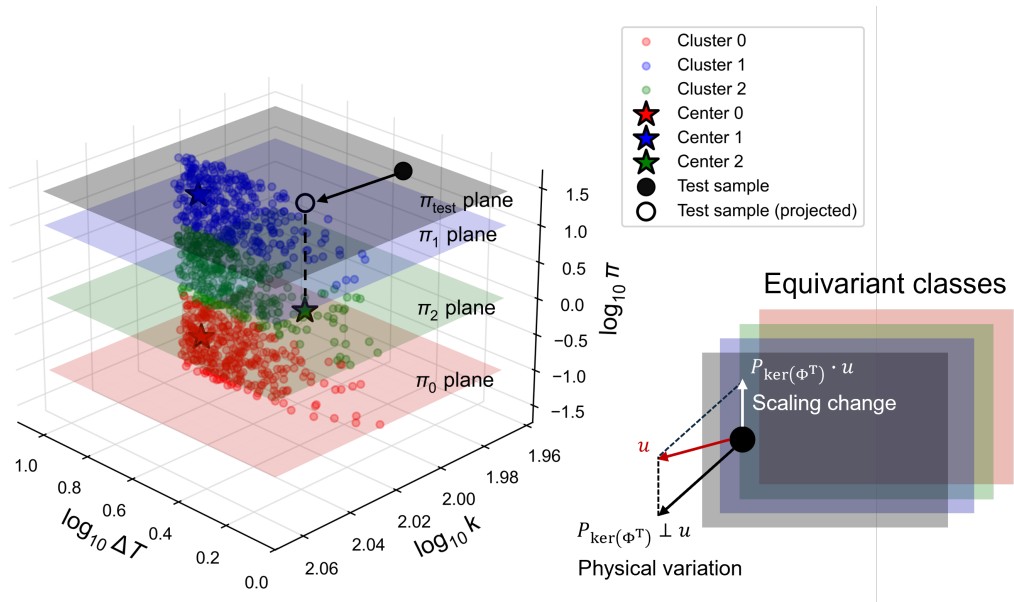

Figure 8: The scheme of clustering projection. Train samples within the same cluster are represented in the same color with centers. The test sample is projected to the nearest center of the cluster while preserving its $\pi_{test}$; it moves on the $\pi_{test}$ plane.

## E EXPERIMENT METRICS

### E.1 MAE AND RMSE EVALUATION

In this experiment, image-wise metric is used for evaluating the MAE, RMSE. For the each image matrix $\{Y_{ij}\}$, MAE can be calculated by

$$\text{MAE}_n = \frac{1}{H \times W} \sum_{i=1}^{H} \sum_{j=1}^{W} \left| \hat{Y}_{ij}^{(n)} - Y_{ij}^{(n)} \right|. \tag{16}$$

Here, $H$ and $W$ denote the image's height and width, respectively. (in this experiment image correspond to color map). using equation 16, image-wise average MAE can be calculated by

$$\text{MAE} = \frac{1}{N} \sum_{i=1}^{N} \text{MAE}_n \tag{17}$$

Additionally, RMSE can be calculated by

$$\text{RMSE}_n = \sqrt{\frac{1}{H \times W} \sum_{i=1}^{H} \sum_{j=1}^{W} \left( \hat{Y}_{ij}^{(n)} - Y_{ij}^{(n)} \right)^2}. \tag{18}$$

from equation 17, image-wise average RMSE can be calculated by

$$\text{RMSE} = \frac{1}{N} \sum_{i=1}^{N} \text{RMSE}_n \tag{19}$$

## F ADDITIONAL PDE EXMAPLE : NAVIER-STOKES EQUATION

We did experiments on Navier-Stokes equation and proved that the projection method also holds for nonlinear problems. The steady governing equations are

$$\rho(\mathbf{u} \cdot \nabla)\mathbf{u} = -\nabla p + \mu \nabla^2 \mathbf{u} + \mathbf{f} \quad \text{in } \Omega, \tag{20}$$

$$\nabla \cdot \mathbf{u} = 0 \quad \text{in } \Omega, \tag{21}$$

where $\mathbf{u} = (u_x, u_y)$ is the velocity, $p$ is the pressure, $\rho$ and $\mu$ are density and dynamic viscosity, and $\mathbf{f}$ is a body force. We consider a domain $\Omega$ with Dirichlet velocity boundary conditions $\mathbf{u} = \mathbf{u}_{\text{bc}}$ on $\partial \Omega_D$. The kinematic viscosity is $\nu = \mu/\rho$, and the Reynolds number is $\mathrm{Re} = \rho \Delta U L / \mu$ where $\Delta U$ is the difference between the maximum and minimum value of $U = \sqrt{u_x^2 + u_y^2}$ on boundaries. This Reynolds number ($\mathrm{Re}$) is used for dimensionless number of the $\pi$-group. We declare the convergence to a steady state when the quantity, $\|\mathbf{u}_x^{t+1} - \mathbf{u}_x^t\|_2 + \|\mathbf{u}_y^{t+1} - \mathbf{u}_y^t\|_2$ (implemented as the sum of the $\ell_2$ norms of $u_x$ and $u_y$ where $\Delta t = 10^{-4}$) drops below a tolerance of $10^{-6}$.

### F.1 NAVIER-STOKES EQUATION (AN IDEAL CASE UNAFFECTED BY EXTERNAL FORCES)

To represent the ideal case, $\mathbf{f}$ is set to 0, assuming that external forcing such as gravity and flow is not present. In this case, the steady governing equations are

$$\rho(\mathbf{u} \cdot \nabla)\mathbf{u} = -\nabla p + \mu \nabla^2 \mathbf{u} \quad \text{in } \Omega, \tag{22}$$

$$\nabla \cdot \mathbf{u} = 0 \quad \text{in } \Omega, \tag{23}$$

For the model training, input data is encoded with 6 channels $\tilde{X} = [\rho, \mu, \mathbf{u}_{\text{bc}}^x, \mathbf{u}_{\text{bc}}^y, dx, dy]$, output data is encoded with 2 channels $[u_x, u_y]$. The scaling coefficients $\alpha, \beta, \gamma$ and $\delta$ are used to scale the parameters while preserving the $\mathrm{Re}$.

$$\rho' = \alpha\rho, \quad \mu' = \beta\mu, \quad \Delta U' = \gamma\Delta U, \quad L' = \delta L \tag{24}$$

For preserving the $\mathrm{Re}$, scaling coefficients should follow the constraint.

$$\mathrm{Re} = \frac{\rho' L' \Delta U'}{\mu'} = \frac{(\alpha\rho)(\delta L)(\gamma\Delta U)}{(\beta\mu)} = \frac{\rho L \Delta U}{\mu} \tag{25}$$

For efficient calculation, log-scale linearization is applied. After the $\pi$-invariant projection, we rescale the test input channel-wise $\tilde{X}^* = \tilde{X} \odot \exp(v^*)$. Then

$$\tilde{X}^* = [\alpha\rho, \beta\mu, \gamma\mathbf{u}_{\text{bc}}^x, \gamma\mathbf{u}_{\text{bc}}^y, \delta dx, \delta dy] \tag{26}$$

By using the $\Delta U$ scale factor $\gamma$, output $\mathbf{u} = (u_x, u_y)$ can be predicted. We observe that applying the $\pi$-invariant projection to the OOD data yields higher performance than making predictions without the projection (see Table 2). *Top-3 best* and *Top-3 worst* OOD cases for Navier-Stokes are shown in Figure 9.

Table 2: Test scores on Navier–Stokes. The best scores are bolded.

| Method | MAE | RMSE | Time |
|---|---|---|---|
| CNN | 0.063±0.004 | 0.100±0.007 | - |
| CNN + Pairwise Projection | **0.016±0.001** | **0.022±0.001** | 27.97±0.9 |
| CNN + $\pi$-uniform + 10-Randoms | 0.018±0.005 | 0.024±0.001 | **0.91±0.03** |
| CNN + $\pi$-uniform + 10-Centroids | 0.017±0.001 | 0.023±0.001 | 0.97±0.02 |
| U-Net | 0.11±0.007 | 0.15±0.01 | - |
| U-Net + Pairwise Projection | **0.02±0.001** | **0.02±0.001** | 28.23±0.76 |
| U-Net + $\pi$-uniform + 10-Randoms | 0.03±0.002 | 0.037±0.002 | **0.92±0.02** |
| U-Net + $\pi$-uniform + 10-Centroids | **0.02±0.001** | 0.03±0.001 | 0.98±0.02 |
| FNO | 0.034±0.003 | 0.044±0.003 | - |
| FNO + Pairwise Projection | **0.013±0.001** | **0.018±0.001** | 28.49±0.77 |
| FNO + $\pi$-uniform + 10-Randoms | 0.016±0.001 | 0.021±0.001 | **1.01±0.04** |
| FNO + $\pi$-uniform + 10-Centroids | 0.014±0.001 | 0.019±0.001 | 1.07±0.05 |

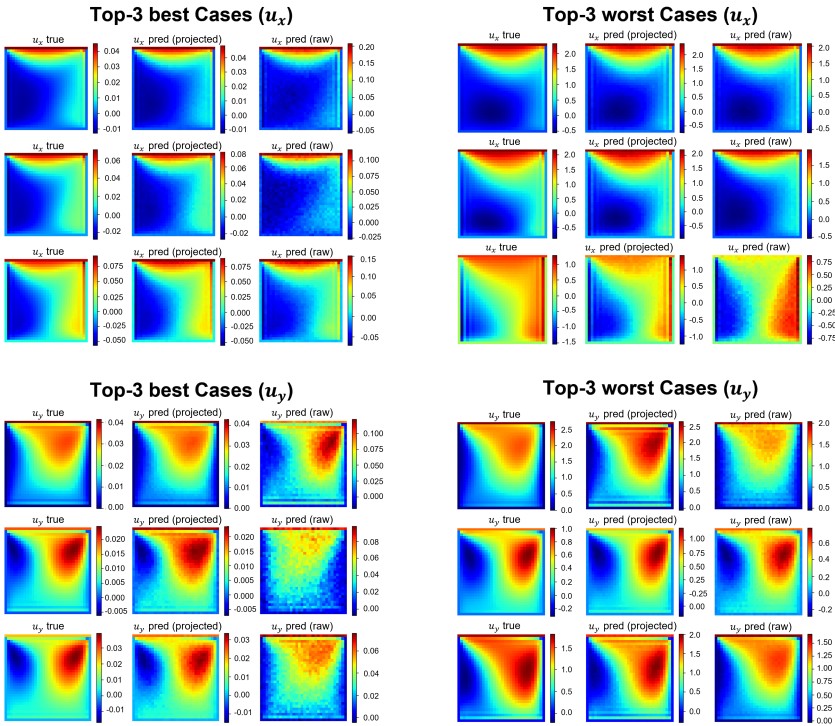

Figure 9: Comparison of ground truths and predictions with *Top-3 best* (a) and *Top-3 worst* (b) cases in Navier-Stokes, under baseline (raw input) vs. Buckingham $\pi$-invariant projection.

## F.2 NAVIER-STOKES EQUATION (AFFECTED BY EXTERNAL FORCES)

The ideal form of the Navier–Stokes equation does not include gravity or any externally imposed flow ($\mathbf{f} = 0$), additional considerations are required when external forcing is present ($\mathbf{f} \neq 0$). However, the $\pi$-groups used in the Buckingham–$\pi$ projection are derived under this ideal assumption ($\mathbf{f} = 0$). Therefore, when $\mathbf{f} \neq 0$, the resulting $\pi$-groups become incomplete.

We also performed experiments to examine how this incompleteness affects the results when such incomplete $\pi$-groups are used. In this setup,

$$\mathbf{f} = \frac{8\mu U}{\rho L^2} \tag{27}$$

is used, where $\mu$ is the viscosity, $U$ is the boundary-condition velocity, $L$ is the characteristic length, and $\rho$ is the density. The characteristic length $L$ was computed as the product of the grid spacing ($\Delta x$) and the resolution.

Table 3: Test Score Comparison for Navier-Stokes ($\mathbf{f} \neq 0$ vs. $\mathbf{f} = 0$) with 10 different test sets. The best scores are bolded.

| Method | Navier-stokes ($\mathbf{f} \neq 0$) | | | Navier-stokes ($\mathbf{f} = 0$) | | |
|---|---|---|---|---|---|---|
| | MAE | RMSE | Time | MAE | RMSE | Time |
| CNN | $0.063 \pm 0.008$ | $0.085 \pm 0.010$ | - | $0.063 \pm 0.004$ | $0.100 \pm 0.007$ | - |
| CNN + Pairwise Projection | $\mathbf{0.029 \pm 0.002}$ | $\mathbf{0.037 \pm 0.003}$ | $28.01 \pm 0.92$ | $\mathbf{0.016 \pm 0.001}$ | $\mathbf{0.022 \pm 0.001}$ | $27.97 \pm 0.9$ |
| U-Net | $0.105 \pm 0.003$ | $0.141 \pm 0.006$ | - | $0.11 \pm 0.007$ | $0.15 \pm 0.01$ | - |
| U-Net + Pairwise Projection | $\mathbf{0.061 \pm 0.002}$ | $\mathbf{0.081 \pm 0.002}$ | $28.19 \pm 0.80$ | $\mathbf{0.02 \pm 0.001}$ | $\mathbf{0.02 \pm 0.001}$ | $28.23 \pm 0.76$ |
| FNO | $0.056 \pm 0.007$ | $0.071 \pm 0.008$ | - | $0.034 \pm 0.003$ | $0.044 \pm 0.003$ | - |
| FNO + Pairwise Projection | $\mathbf{0.025 \pm 0.002}$ | $\mathbf{0.032 \pm 0.002}$ | $28.56 \pm 0.73$ | $\mathbf{0.013 \pm 0.001}$ | $\mathbf{0.018 \pm 0.001}$ | $28.49 \pm 0.77$ |

Table 3 shows that even in the incomplete case ($\mathbf{f} \neq 0$), applying the Buckingham- $\pi$ projection improves prediction accuracy. Although the accuracy is slightly lower compared to the ideal case ($\mathbf{f} = 0$), this result demonstrates that the Buckingham-$\pi$ projection can still enhance prediction performance under incomplete conditions.

## G  OUT-OF DISTRIBUTION DATA IN DIMENSIONLESS SPACE

In this experiment, we evaluate how much the improvement offered by the Buckingham–$\pi$ invariant projection degrades when the dimensionless numbers extracted from the inputs of OOD data lie outside the $\pi$-distribution obtained from the training data, depending on how far they deviate. For the experiment, we use the Reynolds number (Re)—one of the $\pi$-groups commonly appearing in the Navier–Stokes equations—as the domain variable. This choice is motivated by the fact that the flow behavior changes significantly with the Reynolds number (Re), making it a suitable quantity for explaining and interpreting the results of our experiment.

For example, when the Re is below 1, the flow is dominated by viscous effects and typically exhibits Stokes flow. When the Re is in the range of 10–40, attached vortices commonly appear, while for Re greater than 40, the flow is dominated by the Von Kármán vortex street. In our setup, the model is trained only on data with Re in the range 0.04-1.01 (Train data), and tested on three datasets: Test-ID (0.04–1.02), Test-OOD (mid) with a range of 0.32–27.06, and Test-OOD (extreme) with a range of 42.28–63.43 (shown in Figure 10).

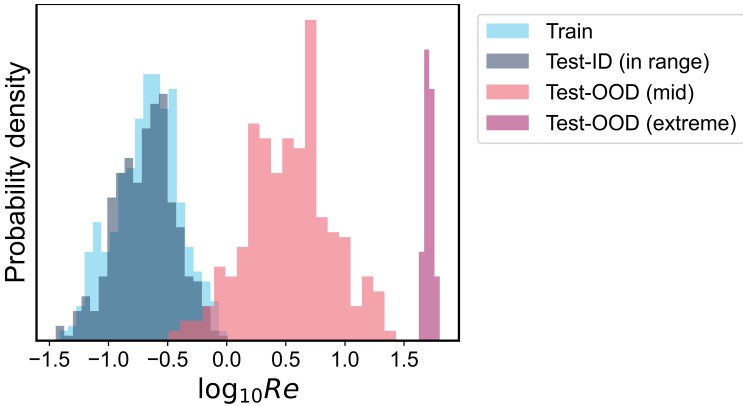

Figure 10: The distribution of $\log_{10}$ Re. Train Re range: 0.04-1.01, Test-ID Re range: 0.04-1.02, Test-OOD (mid) Re range: 0.32-27.06, Test-ID (extreme) Re range: 42.28-64.43.

Table 4: Test scores on Navier-Stokes with 3 different test sets. The best scores are bolded.

| Method | Test-ID (in-range) | | | Test-OOD (mid) | | | Test-OOD (extreme) | | |
|---|---|---|---|---|---|---|---|---|---|
| | MAE | RMSE | R2 | MAE | RMSE | R2 | MAE | RMSE | R2 |
| CNN | 0.02 | 0.03 | 0.02 | 0.20 | 0.28 | 0.12 | 0.38 | 0.59 | -0.04 |
| CNN + Pairwise Projection | **0.005** | **0.009** | **0.82** | **0.08** | **0.13** | **0.76** | **0.34** | **0.52** | **0.18** |
| U-Net | 0.01 | 0.02 | 0.50 | 0.23 | 0.30 | 0.10 | 0.38 | 0.54 | 0.18 |
| U-Net + Pairwise Projection | **0.003** | **0.004** | **0.97** | **0.06** | **0.09** | **0.92** | **0.24** | **0.32** | **0.70** |
| FNO | 0.006 | 0.007 | 0.95 | 0.20 | 0.26 | 0.39 | 0.33 | 0.45 | 0.14 |
| FNO + Pairwise Projection | **0.002** | **0.002** | **0.99** | **0.04** | **0.05** | **0.97** | **0.21** | **0.27** | **0.78** |

Table 4 shows that as the test dataset moves farther away from the training dataset, the prediction accuracy gradually decreases despite the improvements provided by the $\pi$-invariant projection. For the CNN model, the Test-ID dataset—which is similar to the training distribution—achieves an R² score of 82%. However, the score drops to 76% for the Test-OOD (mid) dataset, and further down to 18% for the Test-OOD (extreme) dataset. This trend appears consistently across different models, including U-Net and FNO. Nevertheless, an interesting observation from these results is

that even in cases where the data lie far outside the $\pi$-distribution, using the $\pi$-invariant projection still improves prediction performance compared to making predictions without it. Although the degree of improvement diminishes when the test dataset deviates too far from the training data, the $\pi$-uniform strategy proposed in the paper provides a way to uniformly cover this range. Moreover, in practical settings, the physically meaningful $\pi$-range can be predefined based on prior knowledge from physics and engineering.

## H   ALGORITHM

---

**Algorithm 1** Buckingham-$\pi$-invariant test-time projection

---

1: Train surrogate $f_\theta$ on $\mathcal{D}_{\text{train}} = \{(X_i, Y_i)\}_{i=1}^M$
2: **for** each test pair $(\tilde{X}_i, \tilde{Y}_i) \in \mathcal{D}_{\text{test}}$ **do**          $\triangleright$ OOD test set, $\mathcal{D}_{\text{test}} = \{(\tilde{X}_i, \tilde{Y}_i)\}_{i=1}^M$
3:      $\tilde{z}_i \leftarrow \log \psi(\tilde{X})$                    $\triangleright \psi$=feature extractor
4:      Set $v_i^t = z_i - \tilde{z}$ and decompose $v_i^t$ into $P_\parallel v_i^t$ (intra-class) and $(I - P_\parallel)v_i^t$ (inter-class).
5:      Select the nearest train sample by
6:      $\arg\min_{i=1,\dots,M} \|v_i^t - v_i^*\|_2$ where $v_i^* = P_\parallel v_i^t$
7:      Form projected input $\tilde{X}^* = \tilde{X} \odot \exp(v_{i^*}^*)$; and predict $\tilde{Y}_{\text{scaled}} \leftarrow f_\theta(\tilde{X}^*)$
8:      $\tilde{Y}_{\text{orig}} \leftarrow \text{INVERSESCALE}(\tilde{Y}_{\text{scaled}}, v_{i^*}^*)$          $\triangleright$ e.g., $T = T_{\min} + \gamma^{-1}(T_{\text{scaled}} - T_{\min})$;
9: **end for**

---

