# OpenReview forum: "Buckingham $\pi$-Invariant Test‑Time Projection for Robust PDE Surrogate Modeling"
_ICLR.cc/2026/Conference — ICLR 2026 Poster_

### Official Review · Reviewer_EMmu · 2025-10-28

**Soundness:** 3
**Presentation:** 2
**Contribution:** 2
**Rating:** 4
**Confidence:** 3

**Summary:**

This paper proposes a Buckingham π-Invariant Test-Time Projection method to improve the out-of-distribution (OOD) robustness of PDE surrogate models such as FNOs, U-Nets, and CNNs. The key idea is that many OOD inputs differ only by unit or scale changes that should be physically equivalent under dimensional analysis. The authors therefore apply the Buckingham π theorem to define a log-space pi-group-preserving projection while aligning each test sample with its nearest training sample in parameter space. They proposed a minimization procedure for this alignment. Experiments on steady 2-D thermal conduction and linear elasticity show up to 90% reduction in OOD error with minimal computational overhead.

**Strengths:**

- The paper adapts a century-old but fundamental physical principle (Buckingham π) into a practical, algorithmic test-time procedure for neural surrogates.
- The combination of π-compliant projection and nearest-sample search is a creative way to transfer dimensional invariance into modern ML practice.
- Training-free and model-agnostic: it can wrap around any pretrained surrogate without re-training or altering the loss, which makes it attractive for applied modeling.
- Smooth minimization: the log-space formulation converts multiplicative scaling into a linear subspace problem, solved neatly by an orthogonal projection.
- The idea may stimulate a broader discussion on how physical similarity and scale invariance can be enforced at inference rather than training time.

**Weaknesses:**

- Limited scope of experiments. Only simple, steady, linear PDEs (conduction and elasticity) are tested. It remains unclear whether the method holds for nonlinear, transient, or multi-physics systems.

- Procedure feels overly elaborate for a scaling correction: The projection, clustering, and least-squares steps may appear heavy compared to straightforward normalization or nondimensionalization. There is a lack of exploration of when the procedure was worth, and when it is simply more favorable to cast more training data points.

- Writing and exposition are often opaque: Key transitions between the physical reasoning, log-space math, and algorithmic steps are difficult to follow without prior familiarity.

- No guarantee of a true physical neighbor: The nearest-sample search may project the test case toward an unrelated training sample if the π-space distribution is sparse or multimodal.

- No discussion on mis-specified π-groups: The procedure assumes the chosen invariant is the correct one; the effect of using incomplete or incorrect π-groups is not analyzed.

- Use of mean field values may fail for heterogeneous inputs: Collapsing distributed fields into global means ignores spatial structure, which can distort π values for systems with strong local variability.

**Questions:**

- How sensitive is the method to the choice of π-group? Could the projection degrade performance if irrelevant or redundant groups are used? In many problems the pi groups are actually ratios between problem geometries, and it is unclear how they are to be chosen.

- For heterogeneous domains, can local or hierarchical π values be used instead of global means?

- How would the method behave on nonlinear or transient PDEs (e.g., Navier–Stokes, Burgers’, advection–diffusion)? Since pi scaling is used extensively in computational fluid mechanics, there should be examples from CFD for well-known pi groups

- What is the computational cost for large training sets before centroid reduction, and how stable are results with different cluster seeds?

---

> ### Author Response · Authors · 2025-11-20
> **Rebuttal by Authors (1/5)**
>
> We sincerely appreciate the reviewer’s thoughtful and constructive feedback. Below, we address each comment carefully. Additional experimental results with each question is available at the link below and will be thoroughly incorporated into our revised paper.
>
> ---
>
>
> ****Weaknesses****
>
> **W1**
>
> We did experiments on Navier-Stokes equation and proved that the projection method also holds for nonlinear problems.
>
>
> The results are shown in Table 2 (https://anonymous.4open.science/r/Review_buckingham-EDC1/reviewer%20EMmu/f=0_navier-stokes_result.png).
>
>
> The steady governing equations are
>
>
> $$
> \rho (\mathbf{u} \cdot \nabla)\mathbf{u}
> = -\nabla p + \mu \nabla^2 \mathbf{u} + \mathbf{f}
> \quad \text{in } \Omega,
> $$
>
> $$
> \nabla \cdot \mathbf{u} = 0
> \quad \text{in } \Omega.
> $$
> where $\mathbf u = (u_x,u_y)$ is the velocity, $p$ is the pressure,
> $\rho$ and $\mu$ are density and dynamic viscosity, and $\mathbf f$ is a body force.
> We consider a domain $\Omega$ with Dirichlet velocity boundary conditions
> $\mathbf u = \mathbf u_{\mathrm{bc}}$ on $\partial\Omega_D$.
> The kinematic viscosity is $\nu = \mu/\rho$, and the Reynolds number is
> $\mathrm{Re} = \rho \Delta U L / \mu$ where $\Delta U$ is the
> difference between the maximum and minimum value of $U=\sqrt{u_x^2+u_y^2}$ on boundaries.
> We declare the convergence to a steady state when the quantity,
> $\||\mathbf u_x^{t+1} - \mathbf u_x^{t}\||_2 + \||\mathbf u_y^{t+1} - \mathbf u_y^{t}\||_2$ (implemented as the sum of the
> $\ell_2$ norms of $u_x$ and $u_y$ where $\Delta t=10^{-4}$) drops below a tolerance of $10^{-6}$. You can see the figure of comparison of true and predicted fields for Top-3 best and Top-3 worst cases in Navier-Stokes, under raw input vs. Buckingham $\pi$-invariant projection in the link:
>
>
> https://anonymous.4open.science/r/Review_buckingham-EDC1/reviewer%20EMmu/navier-stokes%20(f=0).png
>
> ---
>
>
> **W2**
>
> Although the proposed procedure may appear conceptually complex, its actual computational cost is extremely small.
>
> K-means clustering is performed only once during training, and the K precomputed centroids are reused at test time with no additional overhead. During test-time projection, the computation reduces to an O(K·N) nearest-centroid search for N test samples, which is a simple vector operation. The subsequent least-squares step involves solving a very small linear system, and both the scaling and inverse-scaling transforms consist of only elementwise multiplications and additions, adding virtually no extra cost.
>
> Overall, the entire process costs less than 1/100 of a single forward pass of most complex neural networks.
>
>
>
> This procedure becomes particularly powerful when predicting out-of-distribution (OOD) data. As shown in the link https://anonymous.4open.science/r/Review_buckingham-EDC1/reviewer%20EMmu/ood_prediction.png, even if substantial computational resources and time are spent generating more training data, the prediction accuracy for OOD samples barely improves. In contrast, applying the Buckingham–
> ${\pi}$ projection enhances prediction accuracy relative to the baseline and exhibits a clear trend of performance improvement as the amount of training data increases.
>
> On the other hand, for in-distribution data, the necessity of using the Buckingham–${\pi}$ projection is lower. In such cases, increasing the number of training samples naturally improves prediction accuracy, making it more advantageous to rely on additional training data rather than applying the projection method.
>
> ---
>
> **W3**
>
> We appreciate this comment and agree that the original exposition could be hard to follow without prior familiarity. We have done several experiments regarding the reviewers' comments and made the explicit step-by-step algorithm which can make the readers easy to follow the process. During the discussion period, we will further polish the wording and incorporate the additional results for readability.
>
> ---
>
>
> **W4**
>
> Depending on the dataset, a truly physically close neighbor is not always guaranteed. However, except for extremely pathological cases, our method still transforms the input in a better direction in most situations.
>
> This relies on the assumption that the surrogate has learned the spatial patterns sufficiently well, and that the mean used in our projection remains a meaningful representative statistic. Nonetheless, as shown in Fig. 5 of the paper, even the worst cases after the $\pi$-projection yield better results than using the raw, unprojected inputs.
>
> In addition, to further mitigate this issue, we introduced the $\pi$-uniform strategy. This approach broadens the $\pi$-distribution and enables coverage of a wider range of $\pi$-values.

---

> ### Author Response · Authors · 2025-11-20
> **Rebuttal by Authors (2/5)**
>
> ****Weaknesses (continued)****
>
> **W5**
>
> The method for defining the $\pi$-groups is derived from classical analyses, and using incorrect $\pi$-groups can significantly degrade prediction accuracy. Moreover, even when an incomplete PDE is used, applying the Buckingham-$\pi$ projection can still be effective.
>
>
> The procedure for defining the $\pi$-groups allows them to be automatically computed via matrix operations based on classical dimensional-analysis theory. Details of this process are provided in https://anonymous.4open.science/r/Review_buckingham-EDC1/reviewer%20EMmu/Buckingham_%CF%80_Invariant_Test_Time_Projection_for_Robust_PDE_Surrogate_Modeling-11-12.pdf which is presented in Appendix A of this paper.
>
>
> If incorrect $\pi$-groups are used, we observe a substantial degradation in prediction performance. Experiments demonstrating this effect are provided at
> https://anonymous.4open.science/r/Review_buckingham-EDC1/reviewer%20EMmu/incorrect.png.
> This degradation occurs because incorrect $\pi$-groups induce improper input scaling, mapping the original input data into a space that no longer reflects the underlying physical characteristics. These observations highlight the importance of defining appropriate $\pi$-groups in the Buckingham-$\pi$ projection.
>
> Additionally, because the ideal form of the Navier–Stokes equation does not include gravity or any externally imposed flow ($\mathbf f=0$), additional considerations are required when external forcing is present ($\mathbf f \neq 0$). However, the $\pi$-groups used in the Buckingham–$\pi$ projection are derived under this ideal assumption ($\mathbf f=0$). Therefore, when $\mathbf f=0$, the resulting $\pi$-groups become incomplete.
> We also performed experiments to examine how this incompleteness affects the results when such incomplete $\pi$-groups are used. Table 3 compares the ideal Navier–Stokes case with
> $\mathbf f=0$  and the incomplete case where a non-zero $\mathbf f$ is given, evaluating both scenarios with and without the Buckingham-${\pi}$ projection (available at: https://anonymous.4open.science/r/Review_buckingham-EDC1/reviewer%20EMmu/compare_ns_equation.png).
>
> In this setup,
> $\mathbf{f} = \frac{8 \mu U}{\rho L^2}$
> , where $\mu$ is the viscosity, $
> U$ is the boundary-condition velocity, $L$ is the characteristic length, and $\rho$ is the density. The characteristic length $L$ was computed as the product of the grid spacing ($\Delta x$) and the resolution. According to the table, even in the incomplete case (non-zero $\mathbf f$), applying the Buckingham-
> ${\pi}$ projection improves prediction accuracy. Although the accuracy is slightly lower compared to the ideal $\mathbf f$ =0 case, this result demonstrates that the Buckingham-${\pi}$ projection can still enhance predictive performance under incomplete conditions. The results with $\mathbf f\neq 0$ are available at https://anonymous.4open.science/r/Review_buckingham-EDC1/reviewer%20EMmu/navier-stokes%20(f!=0).png
>
> **Table 3: Test Score Comparison for Navier-Stokes ($\mathbf{f} \neq 0$ vs. $\mathbf{f} = 0$) with 10 different test sets. The best scores are bolded.**
>
> | **Method** | **MAE** (f $\neq$ 0) | **RMSE** (f $\neq$ 0) | **Time** (f $\neq$ 0) | **MAE** (f = 0) | **RMSE** (f = 0) | **Time** (f = 0) |
> | :--- | :---: | :---: | :---: | :---: | :---: | :---: |
> | CNN | 0.063 $\pm$ 0.008 | 0.085 $\pm$ 0.010 | - | 0.063 $\pm$ 0.004 | 0.100 $\pm$ 0.007 | - |
> | CNN + Pairwise Projection | **0.029 $\pm$ 0.002** | **0.037 $\pm$ 0.003** | 28.01 $\pm$ 0.92 | **0.016 $\pm$ 0.001** | **0.022 $\pm$ 0.001** | 27.97 $\pm$ 0.9 |
> | U-Net | 0.105 $\pm$ 0.003 | 0.141 $\pm$ 0.006 | - | 0.11 $\pm$ 0.007 | 0.15 $\pm$ 0.01 | - |
> | U-Net + Pairwise Projection | **0.061 $\pm$ 0.002** | **0.081 $\pm$ 0.002** | 28.19 $\pm$ 0.80 | **0.02 $\pm$ 0.001** | **0.02 $\pm$ 0.001** | 28.23 $\pm$ 0.76 |
> | FNO | 0.056 $\pm$ 0.007 | 0.071 $\pm$ 0.008 | - | 0.034 $\pm$ 0.003 | 0.044 $\pm$ 0.003 | - |
> | FNO + Pairwise Projection | **0.025 $\pm$ 0.002** | **0.032 $\pm$ 0.002** | 28.56 $\pm$ 0.73 | **0.013 $\pm$ 0.001** | **0.018 $\pm$ 0.001** | 28.49 $\pm$ 0.77 |

---

> ### Author Response · Authors · 2025-11-20
> **Rebuttal by Authors (3/5)**
>
> ****Weaknesses (continued)****
>
> **W5 (continued)**
>
>
>
> **Table 4: Test Score Comparison for thermal conduction (correct $\pi$-groups vs. incorrect $\pi$-group) with 10 different test sets. The best scores are bolded.**
>
> https://anonymous.4open.science/r/Review_buckingham-EDC1/reviewer%20EMmu/incorrect.png
>
> |  | Model type | MAE | RMSE | R2 |
> | :--- | :--- | :---: | :---: | :---: |
> | **Baseline** | CNN | 9.69 $\pm$ 0.49 | 11.08 $\pm$ 0.59 | –4.38 $\pm$ 4.04 |
> | | U-Net | 13.18 $\pm$ 0.34 | 14.80 $\pm$ 0.38 | –2.56 $\pm$ 0.21 |
> | | FNO | 9.77 $\pm$ 1.11 | 11.34 $\pm$ 1.17 | $-1.58 \pm 0.69$ |
> | $\pi_{th} = \frac{qL^{2}}{k\Delta T}$ (correct $\pi$)| CNN | **0.96 $\pm$ 0.07** | **1.25 $\pm$ 0.08** | **0.96 $\pm$ 0.004** |
> | | U-Net | **1.22 $\pm$ 0.13** | **1.65 $\pm$ 0.16** | **0.93 $\pm$ 0.01** |
> | | FNO | **0.87 $\pm$ 0.12** | **1.12 $\pm$ 0.15** | **0.97 $\pm$ 0.007** |
> | $\pi_{th} = \frac{q\Delta T}{kL^{2}}$ (incorrect $\pi$)| CNN | 4.45 $\pm$ 0.63 | 5.58 $\pm$ 0.70 | $-4.86 \pm 5.76$ |
> | | U-Net | 4.90 $\pm$ 0.41 | 6.49 $\pm$ 0.74 | $-2.50 \pm 2.01$ |
> | | FNO | 4.23 $\pm$ 0.60 | 5.33 $\pm$ 0.69 | $-1.27 \pm 1.16$ |
>
> ---
>
>
>
>
> **W6**
>
> There exist various ways to construct a feature extractor for Buckingham–${\pi}$ projection. Among these options, We chose the mean value because it is easy to apply and inherently robust to zero outliers. We confirmed in both the thermal and stress problems—where heterogeneous inputs are provided—that using the mean as a feature extractor is not only robust but also remarkably effective, even when handling discontinuous input terms (e.g.,
> $q$ in the thermal problem).
>
>
> These observations indicate that using the mean as a feature extractor is a reasonable and practical choice for Buckingham–${\pi}$ projection, especially in settings where heterogeneous inputs are provided.
>
>
> Developing an appropriate feature extractor that captures spatial characteristics remains one of our future research directions.
>
> ----
>
>
>
> ****Questions****
>
> **Q1**
>
> *1. How sensitive is the method to the choice of ${\pi}$-group?*
>
>
> Based on the Buckingham $\pi$-theorem, the number of $\pi$ groups is defined as the total number of input and output variables in the given PDE minus the number of independent physical dimensions used by those variables (ex: mass
> $M$, length $L$, time $T$, temperature $\Theta$). Therefore, depending on the PDE, multiple $\pi$-group elements may arise, and ideally, all of them should be considered when performing a Buckingham–$\pi$ projection.
> However, it is also possible to select only a subset of these elements and still achieve a performance level similar to that obtained when using all $\pi$-groups (results are available at: https://anonymous.4open.science/r/Review_buckingham-EDC1/reviewer%20EMmu/Multi-pi%20projection%20result.png). This largely depends on the physical variables that constitute each
> $\pi$-group. Since different
> $\pi$-group elements may be formed from different combinations of physical variables, minor differences in performance can appear depending on the sensitivity of the model to those underlying physical quantities.
>
> **Table 5: Comparison of performance with multi-$\pi$ projection and single $\pi$ projection in stress case with Unet. 10 test sets are used and the best scores are bolded.**
>
> | **Method** | **MAE** | **RMSE** | **R2** |
> | :--- | :---: | :---: | :---: |
> | Single-$\pi$ projection ($fL/E$) | 0.209 $\pm$ 0.009 | 0.379 $\pm$ 0.018 | 0.885 $\pm$ 0.0003 |
> | Multi-$\pi$ projection ($fL/E$, $\sigma/E$, $f\Delta u/E$) | **0.208 $\pm$ 0.011** | **0.378 $\pm$ 0.02** | **0.886 $\pm$ 0.0007** |
>
>
>
> *2. Could the projection degrade performance if irrelevant or redundant groups are used?*
>
>
> For irrelevant groups, whether the projection performance degrades depends on irrelevant term of the PDE. For example, In the case of an ideal PDE with added noise (as in the previously mentioned Navier–Stokes equation with $\mathbf{f} \neq 0$), Buckingham-$\pi$ projection can still provide meaningful performance improvements even when making out-of-distribution (OOD) predictions (https://anonymous.4open.science/r/Review_buckingham-EDC1/reviewer%20EMmu/compare_ns_equation.png). However, compared to the ideal case ($\mathbf{f} = 0$), the prediction accuracy can be slightly lower. If the empirical term
> \textbf{f} accounts for a large portion of the PDE, the improvement in performance may not be as significant as in the ideal scenario.
>
> For redundant cases, note that Buckingham-$\pi$ projection operates as a linear projection in log parameter space. Because of this, using a redundant ${\pi}$-group typically produces results that are nearly identical to those obtained using only the basis ${\pi}$-groups. Although there may be slight differences depending on how strongly the variables included in each group influence the output (i.e., variable sensitivity), these differences were found to be minimal.

---

> ### Author Response · Authors · 2025-11-20
> **Rebuttal by Authors (4/5)**
>
> ****Questions (continued)****
>
> **Q1 (continued)**
>
> *3. In many problems the $\pi$ groups are actually ratios between problem geometries, and it is unclear how they are to be chosen.*
>
>
> The $\pi$-groups are actually ratios between problem geometries, and there exists many possible ways to construct global summaries of the $\pi$ (e.g., using medians, learned feature extractors, and so on). In this case, we chose global $\pi$ summaries using mean values and found that a straightforward mean-based summary worked well. For this initial study, we therefore kept the design minimal, and we leave a more systematic exploration of methods to future work.
>
> ---
>
>
>
> **Q2**
>
> Your suggestion regarding hierarchical or local $\pi$ structures is an interesting idea that we had not considered.
>
> However, extending “local $\pi$” to the extreme—e.g., defining pixel-wise $\pi$ values—immediately introduces structural issues. At the pixel level, reference quantities used in the $\pi$ definition can be zero or extremely small. In heat conduction, for instance, q frequently becomes zero over large regions (as seen in Fig. 2), and if such a term appears in the denominator, the local $\pi$ value collapses to 0 or diverges to $\infty$.
> This type of singularity is inherent to pixel-wise or highly local $\pi$ formulations and is a well-known limitation of such approaches.
>
> That said, your suggestion of “local or hierarchical $\pi$ summaries” is promising because it may retain spatial structure without suffering from these singularities, offering a more robust alternative to purely global summarization. We will consider this direction more carefully in future work.
>
>
> ---
>
>
> **Q3**
>
> As one illustrative example, we conducted experiments using the steady-state Navier–Stokes equations.
>
>
> The results are shown in Table 2 (https://anonymous.4open.science/r/Review_buckingham-EDC1/reviewer%20EMmu/f=0_navier-stokes_result.png).
>
>
> The steady governing equations are
>
>
> $$
> \rho (\mathbf{u} \cdot \nabla)\mathbf{u}
> = -\nabla p + \mu \nabla^2 \mathbf{u} + \mathbf{f}
> \quad \text{in } \Omega,
> $$
>
> $$
> \nabla \cdot \mathbf{u} = 0
> \quad \text{in } \Omega.
> $$
> where $\mathbf u = (u_x,u_y)$ is the velocity, $p$ is the pressure,
> $\rho$ and $\mu$ are density and dynamic viscosity, and $\mathbf f$ is a body force.
> We consider a domain $\Omega$ with Dirichlet velocity boundary conditions
> $\mathbf u = \mathbf u_{\mathrm{bc}}$ on $\partial\Omega_D$.
> The kinematic viscosity is $\nu = \mu/\rho$, and the Reynolds number is
> $\mathrm{Re} = \rho \Delta U L / \mu$ where $\Delta U$ is the
> difference between the maximum and minimum value of $U=\sqrt{u_x^2+u_y^2}$ on boundaries.
> We declare the convergence to a steady state when the quantity,
> $\||\mathbf u_x^{t+1} - \mathbf u_x^{t}\||_2 + \||\mathbf u_y^{t+1} - \mathbf u_y^{t}\||_2$ (implemented as the sum of the
> $\ell_2$ norms of $u_x$ and $u_y$ where $\Delta t=10^{-4}$) drops below a tolerance of $10^{-6}$. You can see the figure of comparison of true and predicted fields for Top-3 best and Top-3 worst cases in Navier-Stokes, under raw input vs. Buckingham $\pi$-invariant projection in the link:
>
> https://anonymous.4open.science/r/Review_buckingham-EDC1/reviewer%20EMmu/navier-stokes%20(f=0).png

---

> ### Author Response · Authors · 2025-11-20
> **Rebuttal by Authors (5/5)**
>
> ****Questions (continued)****
>
>
> **Q4**
>
> The term 'computational cost' refers to the total searching time for the nearest equivalence class to project the test sample, and these values are reported in the Time column of Table 1 (e.g., for thermal case with CNN, the time cost of 100.31$\pm$2.06 seconds reduced to 1.80$\pm$0.02 when using centroid reduction).
> These timings come from an unoptimized implementation and can be further reduced.
>
> In addition, to evaluate stability, we additionally do experiments for the results shown in Table 6 with using 1 test set and 10 different seeds for clustering/random projection (available at: https://anonymous.4open.science/r/Review_buckingham-EDC1/reviewer%20EMmu/clustering_seed.png)
>
> **Table 6: Test scores on thermal and elasticity with 1 test set and 10 different seed for clustering/random projection. The best scores are bolded.**
>
> | Method | MAE (Thermal) | RMSE (Thermal) | Time (Thermal) | MAE (Stress) | RMSE (Stress) | Time (Stress) |
> | :--- | :---: | :---: | :---: | :---: | :---: | :---: |
> | CNN + $\pi$-uniform + 10-Centroids | **1.74 $\pm$ 0.05** | **2.17 $\pm$ 0.06** | 1.79 $\pm$ 0.03 | **0.59 $\pm$ 0.004** | **0.84 $\pm$ 0.005** | 1.34 $\pm$ 0.01 |
> | CNN + $\pi$-uniform + 10-Randoms | 1.86 $\pm$ 0.15 | 2.32 $\pm$ 0.18 | **1.73 $\pm$ 0.04** | 0.64 $\pm$ 0.01 | 0.88 $\pm$ 0.02 | **1.28 $\pm$ 0.008** |
> | U-Net + $\pi$-uniform + 10-Centroids | **1.12 $\pm$ 0.03** | **1.46 $\pm$ 0.02** | 2.26 $\pm$ 0.008 | **0.22 $\pm$ 0.003** | **0.41 $\pm$ 0.004** | 1.55 $\pm$ 0.06 |
> | U-Net + $\pi$-uniform + 10-Randoms | 1.23 $\pm$ 0.08 | 1.60 $\pm$ 0.1 | **2.24 $\pm$ 0.01** | 0.23 $\pm$ 0.03 | 0.42 $\pm$ 0.04 | **1.50 $\pm$ 0.08** |
> | FNO + $\pi$-uniform + 10-Centroids | **1.19 $\pm$ 0.07** | **1.52 $\pm$ 0.08** | 2.32 $\pm$ 0.02 | **0.34 $\pm$ 0.004** | **0.55 $\pm$ 0.004** | 1.51 $\pm$ 0.08 |
> | FNO + $\pi$-uniform + 10-Randoms | 1.32 $\pm$ 0.15 | 1.68 $\pm$ 0.17 | **2.24 $\pm$ 0.02** | 0.36 $\pm$ 0.02 | 0.57 $\pm$ 0.023 | **1.42 $\pm$ 0.09** |

---

> > ### Comment · Reviewer_EMmu · 2025-11-26
> >
> > Thanks for the replies and new results. I will consider revising my score higher after some careful reading. So far it looks good.

---

> > > ### Comment · Reviewer_EMmu · 2025-11-27
> > >
> > > Your training data still has to cover different regimes of flow, right? I suppose one of the pi term is reynolds number, how does the procedure work if the test is far outside the training Re regime? Your paper needs some results on this.

---

> > > > ### Author Response · Authors · 2025-11-29
> > > > **Response to the Reviewer EMmu**
> > > >
> > > > As explicitly stated in Sec.~7.1 (Limitations),
> > > >
> > > >  when a test case lies outside the trained $\pi$-range (``extreme OOD''), our procedure (Eq.~(4)) projects the test input to the *nearest training $\pi$-equivalence class* and rescales.
> > > >
> > > > This stabilizes units/scales but cannot remove a genuine change of physical regime; therefore errors increase, as expected.
> > > > However, we note that the proposed projection method consistently yields significantly better performance than the baseline, on all OOD test sets.
> > > >
> > > >
> > > > In practice, the admissible $\pi$-range is typically known from physics and domain knowledge; we therefore cover that range using the proposed *$\pi$-uniform sampling*.
> > > > To address the comment directly, we generated additional steady-state Navier-Stokes ground truth at higher $Re$ where numerical convergence is challenging. We note that the proposed projection method consistently yields significantly better performance than the baseline, including on all OOD test sets.
> > > >
> > > > The dataset histogram and results are available at:
> > > >
> > > > **1. Dataset distribution:**
> > > >
> > > > https://anonymous.4open.science/r/Review_buckingham-EDC1/reviewer%20EMmu/Re-histogram.png
> > > >
> > > > **2. Results table:**
> > > >
> > > > https://anonymous.4open.science/r/Review_buckingham-EDC1/reviewer%20EMmu/Re-table.png
> > > >
> > > >
> > > >
> > > >
> > > > | Method                       | Test-ID (in-range) MAE | Test-ID (in-range) RMSE | Test-ID (in-range) R2 | Test-OOD (mid) MAE | Test-OOD (mid) RMSE | Test-OOD (mid) R2 | Test-OOD (extreme) MAE | Test-OOD (extreme) RMSE | Test-OOD (extreme) R2 |
> > > > |-----------------------------|-------------------|---------------------|------------------|-------------------|---------------------|------------------|-------------------|---------------------|------------------|
> > > > | CNN                         | 0.02              | 0.03                | 0.02             | 0.20              | 0.28                | 0.12             | 0.38              | 0.59                | -0.04            |
> > > > | CNN + Pairwise Projection   | **0.005**         | **0.009**           | **0.82**         | **0.08**          | **0.13**            | **0.76**         | **0.34**          | **0.52**            | **0.18**         |
> > > > | U-Net                       | 0.01              | 0.02                | 0.50             | 0.23              | 0.30                | 0.10             | 0.38              | 0.54                | 0.18             |
> > > > | U-Net + Pairwise Projection | **0.003**         | **0.004**           | **0.97**         | **0.06**          | **0.09**            | **0.92**         | **0.24**          | **0.32**            | **0.70**         |
> > > > | FNO                         | 0.006             | 0.007               | 0.95             | 0.20              | 0.26                | 0.39             | 0.33              | 0.45                | 0.14             |
> > > > | FNO + Pairwise Projection   | **0.002**         | **0.002**           | **0.99**         | **0.04**          | **0.05**            | **0.97**         | **0.21**          | **0.27**            | **0.78**         |
> > > >
> > > >
> > > > The original training data were drawn from the solver’s stable-convergence regime; the severe convergence difficulties at very high $Re$ indicate these cases are outside the intended steady-state assumptions (i.e., ``extreme OOD''). We will clarify this scope in the paper and place the new figures in the appendix.

---

### Official Review · Reviewer_QjXX · 2025-10-28

**Soundness:** 2
**Presentation:** 1
**Contribution:** 3
**Rating:** 4
**Confidence:** 3

**Summary:**

This paper proposes the use of a test-time projection method in which samples are rescaled to match the scaling of the nearest point in the training data without changing the nondimensional parameters of the system through use of the Buckingham-\pi. The approach is made more efficient by the use of centroids rather than comparisons to the full training data.

**Strengths:**

1. It's a very sensible approach which tackles an important problem.
2. The experiments are informative and show really strong performance.
3. The authors' were clearly conscious of the impact of cost and found an approximation that works well.

Overall, it seems the method is a natural and very promising approach for mitigating OOD performance, at least in the cases where the OOD is due to different dimensional choices could easily happen when applying a pretrained model on new data.

**Weaknesses:**

While there are some unanswered questions and aspects that could be tightened up which I'll list further down, the main reason for my current recommendation is that the presentation could use significant improvement, particularly on writing to a machine learning audience where a many readers will have very little experience with dimensional analysis. I think this can be easily rectified with some restructuring and building out examples better. Here are some concrete issues and suggestions:
1. Many readers who work on neural surrogates are not going to be familiar with dimensional analysis (not arguing that this should be the case, but it is currently true), so when section 3 doesn't explain the basic concept well, subsequent sections will be harder to understand. If you expanded the worked example and using it through each stage of the section to describe what the fields are, what the units are, and then go through the current example showing how to extract $\pi_{th}$ from them would make the statements significantly more concrete and easier to follow.
2. It feels like concepts are often not explained in the place where they are presented. One would expect all of the methods to be explained in section 4 either mathematically or algorithmically, but section 4.4 just explains the uniform strategy as "tunes the dominant scale while others fixed". If this is supposed to be a method that doesn't require users to perform dimensional analysis on their own, how should they determine the dominant scale? How does this make the distribution uniform?
3. Experiments are just equations right now. What types of common applications do these equations represent and why are they interesting test beds? How did you generate the data sets? How are initial conditions generated?

Other issues:
1. Experiments are currently fairly weak. These are two linear problems. It would be more interesting to see if the advantages from a linear projection method also holds for nonlinear problems.
2. What are the test and train distribution of the invariants? For instance, it seems like q and k are shifted in the same pattern which wouldn't necessarily result in disjoint equations. It would be good to highlight where the method would be expected to fail as well.

Minor:
1. B not defined in Theorem 1.
2. Random selection is mentioned, but not included in table 1.

**Questions:**

1. Often, these surrogate methods you're describing are trained on simulation data which is already non-dimensionalized. How would you expect the performance of this method to be affected in this setting where new data is likely using the same characteristic scales? Given the is already dimensionally equivalent, will performance be affected?
2. What's the justification for using the mean in place of characteristic scales? Often in fluids, these scales are relevant to the geometry of some feature in the system. Is this something that can be ignored in the current setting?
3. Could you provide more detail on how the datasets were generated?

---

> ### Author Response · Authors · 2025-11-20
> **Rebuttal by Authors (1/3)**
>
> We sincerely appreciate the reviewer’s thoughtful and constructive feedback. Below, we address each comment carefully. Additional experimental results with each question is available at the link below and will be thoroughly incorporated into our revised paper.
>
> ---
>
>
>
> ****Weaknesses****
>
> **W1**
>
> In the revised manuscript, we added the additional guidance on Appendix A (https://anonymous.4open.science/r/Review_buckingham-EDC1/reviewer%20QjXX/Buckingham_%CF%80_Invariant_Test_Time_Projection_for_Robust_PDE_Surrogate_Modeling-11-12.pdf) that gives units and parameters of each case. We will update the section 3 in main text during the discussion period.
>
> ---
>
>
> **W2**
>
> In our experiments, the dominant parameter is defined as the input parameter whose scale most directly controls the Buckingham $\pi$-group.
>
>
> Concretely, it can be done with SHAP analysis on $\pi$ group for each parameter.
> For example, the dominant parameter $q$ in thermal case shows the highest importance among whole parameters' contribution (48.7\%) for $\pi_{th}$. The result of SHAP analysis is available at the link below:
>
>
> https://anonymous.4open.science/r/Review_buckingham-EDC1/reviewer%20QjXX/dominant_scale.png
>
> Once the dominant parameter is fixed, the $\pi$-uniform strategy does not require any ground truth of PDE solutions. We first sample the input parameters from broad uniform ranges and compute their $\pi$-values. From this $\pi$-distribution, we define a target uniform distribution in $\pi$-space; each training sample adjusts only the dominant parameter (e.g., $q$ for thermal) so that its $\pi$-value matches the target while other inputs fixed. This yields the training set whose induced $\pi$-values are approximately uniform without *implementing the PDE solver*. We will clarify the notion of the dominant parameter and construction of uniform $\pi$ distribution more explicitly in Section 4.4 of the main text during the discussion period.
>
> ---
>
>
> **W3**
>
>
> The stress (elasticity) and thermal conduction PDEs we study are used across a wide range of engineering applications from nm to m scales, where variations in geometry, loading, and material properties frequently lead to OOD conditions in practice.
>
> For thermal conduction, we use the steady‐state heat equation, which is the standard model employed in chip and package thermal management, battery cooling, and power-electronics heat dissipation.
> For stress (elasticity), we use a standard linear elasticity equation that is widely applied to mechanical components, semiconductor packages, MEMS, and civil or structural elements.
> In addition, as part of this rebuttal, we newly include experiments on the Navier–Stokes equations to demonstrate that our method also extends naturally to representative CFD settings.
>
> All datasets are synthetic: all parameters, including boundary and initial conditions, are independently sampled from uniform distributions over physically meaningful ranges for each PDE.
> Coefficient and source fields are generated using Perlin/Simplex libraries—which create natural terrain- or wave-like patterns—together with normalization for scale control under various random seeds.

---

> ### Author Response · Authors · 2025-11-20
> **Rebuttal by Authors (2/3)**
>
> ****Other issues****
>
> **O1**
>
> We did experiments on Navier-Stokes equation and proved that the projection method also holds for nonlinear problems.
>
> The proposed $\pi$-invariant projection is itself independent of linearity; it is just a linear transformation in log-parameter space defined by the dimensional matrix, so the same procedure applies to nonlinear PDEs as well.
> The results are shown in Table 2 (https://anonymous.4open.science/r/Review_buckingham-EDC1/reviewer%20QjXX/f=0_navier-stokes_result.png).
>
> The steady governing equations are
>
>
> $$
> \rho (\mathbf{u} \cdot \nabla)\mathbf{u}
> = -\nabla p + \mu \nabla^2 \mathbf{u}
> \quad \text{in } \Omega,
> $$
>
> $$
> \nabla \cdot \mathbf{u} = 0
> \quad \text{in } \Omega.
> $$
> where $\mathbf u = (u_x,u_y)$ is the velocity, $p$ is the pressure,
> $\rho$ and $\mu$ are density and dynamic viscosity.
> We consider a domain $\Omega$ with Dirichlet velocity boundary conditions
> $\mathbf u = \mathbf u_{\mathrm{bc}}$ on $\partial\Omega_D$.
> The kinematic viscosity is $\nu = \mu/\rho$, and the Reynolds number is
> $\mathrm{Re} = \rho \Delta U L / \mu$ where $\Delta U$ is the
> difference between the maximum and minimum value of $U=\sqrt{u_x^2+u_y^2}$ on boundaries.
> We declare the convergence to a steady state when the quantity,
> $\||\mathbf u_x^{t+1} - \mathbf u_x^{t}\||_2 + \||\mathbf u_y^{t+1} - \mathbf u_y^{t}\||_2$ (implemented as the sum of the
> $\ell_2$ norms of $u_x$ and $u_y$ where $\Delta t=10^{-4}$) drops below a tolerance of $10^{-6}$. You can see the figure of comparison of true and predicted fields for Top-3 best and Top-3 worst cases in Navier-Stokes, under raw input vs. Buckingham $\pi$-invariant projection in the link:
>
> https://anonymous.4open.science/r/Review_buckingham-EDC1/reviewer%20QjXX/navier-stokes%20(f=0).png
>
> **Table 2. Test scores on Navier–Stokes (best scores in bold).**
>
> | Method                                   | MAE                 | RMSE                | Time         |
> |------------------------------------------|---------------------|---------------------|------------------|
> | CNN                                      | 0.063 ± 0.004       | 0.100 ± 0.007       | –                |
> | CNN + Pairwise projection                | **0.016 ± 0.001**   | **0.022 ± 0.001**   | 27.97 ± 0.90     |
> | CNN + $\pi$-uniform + 10-centroids      | 0.017 ± 0.001       | 0.023 ± 0.001       | 0.97 ± 0.02      |
> | CNN + $\pi$-uniform + 10-randoms        | 0.018 ± 0.005       | 0.024 ± 0.001       | **0.91 ± 0.03**  |
> | U-Net                                    | 0.11 ± 0.007        | 0.15 ± 0.01         | –                |
> | U-Net + Pairwise projection             | **0.02 ± 0.001**        | **0.02 ± 0.001**        | 28.23 ± 0.76     |
> | U-Net + $\pi$-uniform + 10-centroids    | **0.02 ± 0.001**    | 0.03 ± 0.001    | 0.98 ± 0.02      |
> | U-Net + $\pi$-uniform + 10-randoms      | 0.03 ± 0.002        | 0.037 ± 0.002       | **0.92 ± 0.02**  |
> | FNO                                      | 0.034 ± 0.003       | 0.044 ± 0.003       | –                |
> | FNO + Pairwise projection               | **0.013 ± 0.001**   | **0.018 ± 0.001**   | 28.49 ± 0.77     |
> | FNO + $\pi$-uniform + 10-centroids      | 0.014 ± 0.001       | 0.019 ± 0.001       | 1.07 ± 0.05      |
> | FNO + $\pi$-uniform + 10-randoms        | 0.016 ± 0.001       | 0.021 ± 0.001       | **1.01 ± 0.04**  |
>
> ---
>
>
> **O2**
>
> We have added the train and test distribution of the invariants in the histogram plot (Figure 4 in the main text).
>
>
> As shown in Figure 4, the $\pi$-distributions of train and test are not exactly the same; the test $\pi$-values are not a simple shifted copy of the train distribution. Our projection method operates only on global scalar summaries (e.g., means of $k$, $q$); it does not modify the spatial patterns of the fields. Spatial structure (e.g., different patterns and min/max ratios in each image) is handled entirely by the surrogate, which has learned these patterns.
>
>
> The method is expected to degrade when test samples have $\pi$-values far outside the training range, or when the physics truly changes.
>
> ----
>
> ****Minor****
>
> **M1**
>
> We revised the manuscript. $B$ refers to the number of base units for dimension matrix $D$.
>
> ---
>
>
> **M2**
>
>  We revised the Table 1 in the main text including the performance of random projection.

---

> ### Author Response · Authors · 2025-11-20
> **Rebuttal by Authors (3/3)**
>
> ****Questions****
>
> **Q1**
>
> Our method is unaffected.
>
>
>
>
> Even if the inputs are pre-scaled as $q' = q / q_{\mathrm{ref}}\$, the Buckingham-$\pi$ groups are preserved up to a constant factor, because they depend only on relative ratios. For the thermal case, we can define a $\pi$-group with non-dimensionalized parameters as $\pi_{th}'$ = $C\pi_{th}\$ where $\pi_{th}=qL^2/k\Delta T$ (with dimensional parameters), $C = k_{ref}$$\Delta T_{ref}/q_{ref}$$L^2_{ref}$ for dataset-wide constant $C$ when each variable is divided by a fixed reference value. Thus the geometry of the $\pi$-space is unchanged ($\log\pi'=\log(C\pi)=\log\pi+\log C$ ); so train and test cases that share the same $\pi$-groups correspond to the same unique solution.
>
> In other words, even if the absolute values of the Buckingham-$\pi$ terms change after dividing by a constant scale, the solution is unaffected because the $\pi$-space is determined only by their relative ratios.
>
>
> The only caveat is that this invariance assumes linear rescaling (simple division by reference values). If the so-called ``non-dimensionalization'' instead uses nonlinear transforms such as standard normalization, the $\pi$-groups are no longer preserved in general, and the projection would not behave as an exact identity in that case.
>
> ---
>
>
> **Q2**
>
> We use the mean because it provides a simple reference that reasonably represents the spatial pattern and remains numerically stable for log-space projection.
>
> As discussed above, the absolute scaling of this reference is irrelevant. The mean captures the central tendency of the spatial pattern and, importantly, never becomes zero or divergent in our setting, making it a robust representative value for an *n*-dimensional field. From the same perspective, using the median yields almost identical performance in practice.
>
> However, other possible choices, for instance the geometric mean, can become zero or even undefined when the field contains zero or negative values, which breaks the log-space projection.
>
> ---
>
>
> **Q3**
>
> The datasets were generated considering physically reasonable ranges of each parameter.
>
>
> For each PDE system, we first sample training inputs and base test inputs by drawing every parameter from a uniform distribution over a physically plausible range, using different random seeds for train and base sets. To generate OOD test samples, we then draw $\pi$-preserving scaling coefficients from uniform distributions and apply them to these base inputs. The coefficient ranges are chosen so that all scaled parameters remain within physically plausible and numerically stable regimes (e.g., initial boundary conditions $T_{bc}$ in test set are in range of [0 $K$, 890 $K$]).

---

### Official Review · Reviewer_pqY4 · 2025-10-28

**Soundness:** 4
**Presentation:** 3
**Contribution:** 3
**Rating:** 8
**Confidence:** 4

**Summary:**

This paper focuses on mitigating out-of-distribution (OoD) inference of neural operators. The method is based on Buckingham $\pi$-theorem, which decomposes the entire spaces into two parts: the null space $\ker(\Phi^T)$ and the component perpendicular to $\ker(\Phi^T)$, where $\Phi=[\phi^{(1)}\cdots\phi^{(p-r)}]$ stacks the null-space bases vectors (see Theorem 1). Thus, data can be transformed or projected to a point that is the closest to training data while preserving $\pi$. Experiments demonstrate effectiveness on OoD data that is superior than baselines.

**Strengths:**

**Originality** is high. The method leverages Buckingham $\pi$-theorem and addresses OoD problem innovatively from a structured perspective, i.e., data space can be decomposed into equivalence classes generated by training data $(X_i, Y_i)$.

**Clarity** is good in general except for some minor issue (see weakness). Diagram and figures are illustrative and helpful.

**Weaknesses:**

Regarding **Clarity**:
1. An algorithm that summarizes all the procedure can be helpful to readers. Especially, corresponding to predict & inverse in Fig. 3. How to do inverse in general?
2. Above equation (11), should $\tilde{X}^{\star}$ be $\tilde{X}^{*}$ as eq. (11)?
3. In Fig. 1, what do purple circles and yellow circles stand for?

Reference:
1. How is your work related to Lie point symmetry [1]?

Limitation:
1. Can your method be applied to irregular mesh grids? All baseline models in your paper, CNN, U-Net and FNO, can only be applied on uniform grids. Is there such limitation for your method?

[1] Brandstetter, J., Welling, M., & Worrall, D. E. (2022, June). Lie point symmetry data augmentation for neural pde solvers. In International Conference on Machine Learning (pp. 2241-2256). PMLR.

**Questions:**

See weakness.

---

> ### Author Response · Authors · 2025-11-20
> **Rebuttal by Authors**
>
> We sincerely appreciate the reviewer’s thoughtful and constructive feedback. Below, we address each comment carefully. Additional experimental results with each question is available at the link below and will be thoroughly incorporated into our revised paper.
>
> ---
>
>
> ****Weaknesses for clarity****
>
>
> **W1**
>
> We agree that an explicit algorithm would improve readability, and we will include the pseudocode shown in https://anonymous.4open.science/r/Review_buckingham-EDC1/reviewer%20pqY4/algorithm.png
>
>
> After identifying the nearest training sample in $\pi$-space, we solve for the $\pi$-preserving scaling coefficients that map the test input onto that training sample. These coefficients define a scaling transform $g$ on the inputs, and we construct the projected input $\tilde X^{*}$ = $g$ $\cdot$ $\tilde X$.
>
> Evaluating the surrogate at $\tilde X^{*}$ yields a prediction in the scaled coordinates, which we denote by $\tilde Y_{\text{scaled}}$.
>
>
> In general, we analytically derive how the true solution changes under the same scaling $g$, i.e., a simple linear relation
> $$
> \tilde Y_{\mathrm{scaled}} = \rho(g)\tilde Y_{\mathrm{orig}},
> $$
> where $\rho(g)$ is an output-side scaling operator (often a single scalar factor for our scalar fields). At test time we therefore recover the prediction in the original physical units by applying the inverse transform
> $$
> \tilde Y_{\mathrm{orig}} = \rho(g)^{-1}\tilde Y_{\mathrm{scaled}}.
> $$
>
> ---
>
>
> **W2**
>
>
> We have corrected the typo. Thank you for your kindness.
>
> ---
>
>
>
> **W3**
>
>  In Fig.1, the purple circles and yellow circles stand for the train samples on each $\pi$-equivalence class; meaning that some $\pi$-equivalence classes can have multiple samples. We will add this information to the Fig. 1 caption during the discussion period.
>
> ---
>
>
> ****Weaknesses for Reference****
>
>
> **W1**
>
>
> The classical Buckingham–$\pi$ theorem can be viewed as a special case of a Lie point symmetry under a scaling group: the associated scaling transformations form a Lie group, and the orbits of this group coincide with the $\pi$–equivalence classes used in our method. In this sense, our work is conceptually related to [1], which also exploits Lie point symmetries of PDEs.
>
> Brandstetter et al. [1] assume that the full Lie point symmetry group of a given PDE is known analytically and use it to perform exact data augmentation, thereby reducing the sample complexity of neural PDE solvers. The commonalities with our approach are that both (i) leverage Lie-type symmetries and (ii) operate over equivalence classes induced by those symmetries. However, the roles are different: [1] is a *pre-training* procedure for data augmentation that requires precise knowledge of the equation and its symmetry group, whereas our Buckingham–$\pi$ projection is a *post-training* test-time adaptation step that selects a representative within a $\pi$–equivalence class and only requires the dimensional matrix (not the full PDE or its symmetry algebra).
>
> The two approaches are complementary, but Lie point symmetry relies on expanding equivalent samples, increasing the training burden. In contrast, our method projects test inputs onto an equivalent sample without any additional training. Even if Lie symmetry could generate infinitely many equivalent samples and reach the same ideal performance, our approach is more efficient in terms of cost.
>
> ---
>
> ****Weaknesses for Limitation****
>
> **W1**
>
> Our projection method itself is not restricted to uniform grids.
>
>
> Our current experiments use CNN/U-Net/FNO surrogates on uniform grids, so we do not empirically claim support for irregular meshes. However, the Buckingham $\pi$-invariant projection itself only rescales physical parameters and characteristic length scales and is agnostic to the underlying mesh topology. Since we already observe robust behavior under highly heterogeneous coefficient fields, we expect the same projection to extend naturally to neural surrogates defined on unstructured meshes, which we leave as future work.

---

### Official Review · Reviewer_VJx6 · 2025-11-04

**Soundness:** 3
**Presentation:** 2
**Contribution:** 2
**Rating:** 2
**Confidence:** 3

**Summary:**

The paper proposes a projection method, called \pi-invariant test-time projection, which can cluster training and test data points into to dimensionless groups. Such group is invariant to the unit scales, and can help separate data points with different physical behaviors.  According to the paper, this can help reduce the prediction performance for OOD samples.

**Strengths:**

1. The introduction of Buckingham \pi-invariant is interesting and new to the CS/ML community --- though this might be an old concept in computational physics/applied math.
2. The idea of clustering data points according to different behaviors without being influenced by units/scale changes is interesting and has pontential to enhance the generalization of current ML surrogate models.

**Weaknesses:**

1. Clarity. This might be the biggest issue --- the paper does not explains clearly how the proposed projection method is integrated into the training/testing pipeline of neural operators to enhance OOD prediction. Throughout, the paper focuses on how to do projection and clustering. but what to do with NO training/testing? Will you train a different NO for each cluster, and then use the test-time projection for each test example to dynamically determine which NO should be used to predict? Or will you compute a soft cluster membership of each cluster, and then perform a mixture of predictions? Intuitively, there can be many ways of integrating the proposed method with NO training/testing or even data preparation/acquisition. However, the part is significantly lacking and it is hard to understand how the improvement in experimental part is obtained.

2. The OOD problem mentioned in this paper is a bit different from commonly used settings. Regarding OOD, we will first change the distribution of the input to neural operators, rather than assume fundamental change of physical behaviors. In fact, an ML surrogate (e.g., NO) is typically used to capture one type of physical behaviors under various scenarios. I am not sure if expanding the scope to make a surrogate model account for several different physics is appropriate or feasible. At least, the paper should clarify its own meaning of OOD, difference/connection with settings in prior works.

3. The experimental results are limited. Only on two systems is not sufficient in this community. Also, there is no standard deviation in Table 1, making it hard to conclude the significance of improvement.

**Questions:**

see above

---

> ### Author Response · Authors · 2025-11-20
> **Rebuttal by Authors (1/2)**
>
> We sincerely appreciate the reviewer’s thoughtful and constructive feedback. Below, we address each comment carefully. Additional experimental results with each question is available at the link below and will be thoroughly incorporated into our revised paper.
>
> ---
>
>
>
> ****Weaknesses****
>
> **W1**
>
>
> Our method does not train different neural operators per cluster, nor does it use a mixture of predictions across clusters. For each PDE (thermal and stress), we always train a single neural network in a completely standard way; the proposed Buckingham $\pi$ projection is applied only at test time to the inputs, and clustering is used only to organize the training set into similar $\pi$-equivalence classes for projection speed. The overall flow is summarized as follows:
>
>
>
> 1. Train a neural network model for each PDE in a standard way.
>
>
>
> 2. $\pi$-invariant projection for each test sample to the nearest train sample; input parameters are scaled.
>
>
>
> 3. Prediction with projected inputs.
>
>
>
> 4. Inverse scaling of the prediction based on the $\pi$-invariant scaling.
>
>
>
> Thus, the improvement reported in the experiments comes purely from adapting OOD test inputs by $\pi$-invariant projection (from OOD to In-distribution), not from training different models per cluster or from any mixture of predictions. Accordingly, we will revise *Method* to clarify:
> *We train a single surrogate; at test time we apply a Buckingham--$\pi$-preserving
> projection to inputs. K-means is used only to select $K$ representative centroids for
> the projection; no per-cluster models or mixtures are trained.*
>
> ---
>
>
>
> **W2**
>
>
> Our OOD setting is entirely conventional; what we call OOD corresponds to the standard covariate shift scenario where the input distribution differs between training and test ($P_{\mathrm{train}}(X)\neq P_{\mathrm{test}}(X)$), typically through more extreme scales or ranges of physical parameters.
>
> In other words, what we call OOD corresponds to the case $P_{\mathrm{train}}(X)\neq P_{\mathrm{test}}(X)$ with the same underlying PDE and solution operator  $P_{\mathrm{train}}(Y\mid X)=P_{\mathrm{test}}(Y\mid X)$.  Our intention is not to train a single surrogate that covers several different physics. We do not change the underlying physics. For each task in the paper, the governing PDEs are fixed. In a nutshell, the ‘OOD’ always refers to the case where the governing PDE is fixed for each task, but the distribution of dimensional input parameters differs from training, most notably through changes in their overall scale or range. In this setting, our contribution is for adapting OOD test inputs by $\pi$-invariant projection process.

---

> ### Author Response · Authors · 2025-11-20
> **Rebuttal by Authors (2/2)**
>
> ****Weaknesses (continued)****
>
>
> **W3**
>
>
> We added the standard deviation in the paper. To address the concern about limited experimental coverage, we do experiments on Navier-Stokes equation and the results are shown in Table 2 (https://anonymous.4open.science/r/Review_buckingham-EDC1/reviewer%20Vjx6/f=0_navier-stokes_result.png).
> The steady governing equations are
>
>
> $$
> \rho (\mathbf{u} \cdot \nabla)\mathbf{u}
> = -\nabla p + \mu \nabla^2 \mathbf{u}
> \quad \text{in } \Omega,
> $$
>
> $$
> \nabla \cdot \mathbf{u} = 0
> \quad \text{in } \Omega.
> $$
> where $\mathbf u = (u_x,u_y)$ is the velocity, $p$ is the pressure,
> $\rho$ and $\mu$ are density and dynamic viscosity.
> We consider a domain $\Omega$ with Dirichlet velocity boundary conditions
> $\mathbf u = \mathbf u_{\mathrm{bc}}$ on $\partial\Omega_D$.
> The kinematic viscosity is $\nu = \mu/\rho$, and the Reynolds number is
> $\mathrm{Re} = \rho \Delta U L / \mu$ where $\Delta U$ is the
> difference between the maximum and minimum value of $U=\sqrt{u_x^2+u_y^2}$ on boundaries.
> We declare the convergence to a steady state when the quantity,
> $\||\mathbf u_x^{t+1} - \mathbf u_x^{t}\||_2 + \||\mathbf u_y^{t+1} - \mathbf u_y^{t}\||_2$ (implemented as the sum of the
> $\ell_2$ norms of $u_x$ and $u_y$ where $\Delta t=10^{-4}$) drops below a tolerance of $10^{-6}$. You can see the figure of comparison of true and predicted fields for Top-3 best and Top-3 worst cases in Navier-Stokes, under raw input vs. Buckingham $\pi$-invariant projection in the link:
>
> https://anonymous.4open.science/r/Review_buckingham-EDC1/reviewer%20Vjx6/navier-stokes%20(f=0).png
>
> **Table 2. Test scores on Navier–Stokes (best scores in bold).**
>
> | Method                                   | MAE                 | RMSE                | Time         |
> |------------------------------------------|---------------------|---------------------|------------------|
> | CNN                                      | 0.063 ± 0.004       | 0.100 ± 0.007       | –                |
> | CNN + Pairwise projection                | **0.016 ± 0.001**   | **0.022 ± 0.001**   | 27.97 ± 0.90     |
> | CNN + $\pi$-uniform + 10-centroids      | 0.017 ± 0.001       | 0.023 ± 0.001       | 0.97 ± 0.02      |
> | CNN + $\pi$-uniform + 10-randoms        | 0.018 ± 0.005       | 0.024 ± 0.001       | **0.91 ± 0.03**  |
> | U-Net                                    | 0.11 ± 0.007        | 0.15 ± 0.01         | –                |
> | U-Net + Pairwise projection             | **0.02 ± 0.001**        | **0.02 ± 0.001**        | 28.23 ± 0.76     |
> | U-Net + $\pi$-uniform + 10-centroids    | **0.02 ± 0.001**    | 0.03 ± 0.001    | 0.98 ± 0.02      |
> | U-Net + $\pi$-uniform + 10-randoms      | 0.03 ± 0.002        | 0.037 ± 0.002       | **0.92 ± 0.02**  |
> | FNO                                      | 0.034 ± 0.003       | 0.044 ± 0.003       | –                |
> | FNO + Pairwise projection               | **0.013 ± 0.001**   | **0.018 ± 0.001**   | 28.49 ± 0.77     |
> | FNO + $\pi$-uniform + 10-centroids      | 0.014 ± 0.001       | 0.019 ± 0.001       | 1.07 ± 0.05      |
> | FNO + $\pi$-uniform + 10-randoms        | 0.016 ± 0.001       | 0.021 ± 0.001       | **1.01 ± 0.04**  |

---

### Author Response · Authors · 2025-11-30
**Common response to all reviewers**

Thanks again to all the reviewers for engaging thoughtfully during the discussion period.


The comments raised many interesting points and have greatly improved the paper.

We have revised the paper in response to these comments, including

**1. Clarifications in Section 3**

**2. Revised the sequence of previous sections (Sec. 4.3-4.5) to improve clarity and presentation**

**3. Additional details in Section 4.3 ("Offline Preparation: $\pi$-uniform strategy & centroid reduction")**

**4. Add the experiment sequence in the beginning of Section 5**

**5. New experimental results in Appendix F and G.**

**6. Algorithm for Buckingham $\pi$-invariant projection in Appendix H.**

We hope these revisions address the concerns and make the work more accessible.

---

### Author Response · Authors · 2025-12-03
**Dear Program Chairs, Area Chair, and Reviewers**

We would first like to express our sincere gratitude to the four reviewers for their thoughtful and in-depth comments on our submission. We also extend our deep appreciation to the Area Chair for taking the time to conduct an additional evaluation of our paper.

The reason we are submitting this final comment is to briefly highlight where the novelty of our work—*“Buckingham $\pi$-Invariant Test-Time Projection for Robust PDE Surrogate Modeling”*—lies, and to clarify why we believe it meets the acceptance criteria for ICLR 2026. A concise summary is provided below.




>**1. Novelty and conceptual significance**

We demonstrate that our method maintains in-distribution–level prediction accuracy even for OOD data exhibiting tens to hundreds of times differences in scale—arising from variations in source terms, material properties, and grid lengths.

This robustness is validated not only on thermal conduction and stress models, but also on nonlinear PDEs such as the Navier–Stokes equations, as shown in our rebuttal-phase experiments.

In particular, for the Navier–Stokes system, we show that the Buckingham-$\pi$ projection yields performance improvements even under non-ideal conditions influenced by external perturbations. Furthermore, we confirm that these gains hold regardless of the choice of PDE surrogate model; our method consistently improves performance across CNNs, U-Nets, and FNO-based solvers.


>**2. Model architecture and robustness to PDE data**

Our study extends this line of research by leveraging dimensionless variables derived from the $\pi$-groups to accurately predict full fields beyond a single scalar quantity which has not been explored previously.

Moreover, our method significantly reduces inference cost by combining the $\pi$-uniform strategy with a centroid-reduction procedure based on the K-means algorithm.

Since such OOD challenges arising from unit and scale shifts were not addressed in earlier works, including the FNO paper published at ICLR, we believe our contributions offer substantial novelty that merits publication at ICLR.


FNO (ICLR 2021) : https://arxiv.org/pdf/2010.08895


>**3. Positive reactions from the reviewers**

During the rebuttal period, the reviewer EMmu explicitly indicated a positive intention to raise the score.

We have also diligently incorporated all comments and concerns raised by the reviewers, and **these revisions are highlighted in blue in the manuscript**. Had the review process proceeded without interruption, we believe there was a clear possibility that other reviewers might likewise have increased their scores. This is because the additional experiments and analyses we conducted in response to the reviewers’ thoughtful comments have substantially strengthened the paper, as reflected in both the main text and the supplementary material. We kindly ask the Area Chair to take these improvements into account while evaluating our submission.


>**4. Rebuttal conducted based on extensive experiments**

We believe that the conducted additional experiments we carefully addressed could meaningfully affect the reviewers' scores to be more positive.

Also, the results of these experiments were incorporated into the paper during the rebuttal period, and they could meaningfully affect the reviewers’ scores. We respectfully ask the Area Chairs to consider these findings when evaluating our submission, referring to both the anonymous link and the experiments presented in the main text.

We sincerely appreciate the thorough review and the time dedicated by the reviewers throughout the evaluation process.


*Sincerely,*


*The Authors*

---

### Meta-Review · Area_Chair_CEod · 2026-01-07

**Summary:**

The paper proposes a test-time adaptation scheme for PDE surrogates using Buckingham π invariants: it projects each test input (in log-parameter space) onto a nearby training point while preserving dimensionless groups, utilizing centroid search to reduce computational cost. Reviewers found the idea promising for a realistic OOD mode (unit/scale shifts), but raised concerns about (i) unclear pipeline integration and lack of calrity for an ML audience, (ii) initially limited experiments and missing variance numbers, and (iii) unclear limits of applicability (e.g., extreme regime shifts, π-group misspecification).

**Reviewer Concerns:**

The authors clarified that training remains unchanged (one standard surrogate per PDE) and π-projection is applied only at test time (without per-cluster models or mixtures). They also committed to adding pseudocode and providing a clearer explanation of scaling/inverse-scaling. They also clarified OOD as a standard covariate shift under fixed physics.

They partially addressed experimental concerns by adding standard deviations and including Navier-Stokes results to demonstrate nonlinearity, along with added discussion on computational cost (centroids) and stability across clustering seeds. They also explicitly acknowledged failure modes when test cases fall far outside the training π-range and added supporting results.

Generalization remains only partially addressed: much of the evaluation is still steady and relatively constrained, and sensitivity to π-group choices and reliance on global summaries (means) for heterogeneous fields is not fully resolved (mostly deferred as future work). The paper would also benefit from clearer guidance on when the method adds value.

**Reviewer Scores:**

The initially negative review would likely move upward to a borderline rating after the rebuttal clarified the pipeline/OOD definition and expanded on the experiments. The strongly positive review would likely remain positive since their issues were mainly minor clarity/positioning points that were addressed. The two borderline reviews would likely increase; one explicitly indicated openness to raising the score after the new results, and the authors added “extreme regime” evidence and limitation framing supports a shift into weak acceptance if integrated cleanly.

---

### Decision · Program_Chairs · 2026-01-26

Accept (Poster)